# LC-Opt: Benchmarking Reinforcement Learning and Agentic AI for End-to-End Liquid Cooling Optimization in Data Centers

**Avisek Naug**[1][†], **Antonio Guillen**[1][†], **Vineet Kumar**[2], **Scott Greenwood**[2], **Wesley Brewer**[2],
**Sahand Ghorbanpour**[1], **Ashwin Ramesh Babu**[1], **Vineet Gundecha**[1],
**Ricardo Luna Gutierrez**[1], **Soumyendu Sarkar**[1][†][*]

[1]Hewlett Packard Enterprise, [2]Oak Ridge National Laboratory

{avisek.naug, antonio.guillen, sahand.ghorbanpour, ashwin.ramesh-babu,
vineet.gundecha, rluna, soumyendu.sarkar}@hpe.com
{kumarv, greenwoodms, brewerwh}@ornl.gov

## Abstract

Liquid cooling is critical for thermal management in high-density data centers with the rising AI workloads. However, machine learning-based controllers are essential to unlock greater energy efficiency and reliability, promoting sustainability. We present LC-Opt, a Sustainable Liquid Cooling (LC) benchmark environment, for reinforcement learning (RL) control strategies in energy-efficient liquid cooling of high-performance computing (HPC) systems. Built on the baseline of a high-fidelity digital twin of Oak Ridge National Lab's Frontier Supercomputer cooling system, LC-Opt provides detailed Modelica-based end-to-end models spanning site-level cooling towers to data center cabinets and server blade groups. RL agents optimize critical thermal controls like liquid supply temperature, flow rate, and granular valve actuation at the IT cabinet level, as well as cooling tower (CT) setpoints through a Gymnasium interface, with dynamic changes in workloads. This environment creates a multi-objective real-time optimization challenge balancing local thermal regulation and global energy efficiency, and also supports additional components like a heat recovery unit (HRU). We benchmark centralized and decentralized multi-agent RL approaches, demonstrate policy distillation into decision and regression trees for interpretable control, and explore LLM-based methods that explain control actions in natural language through an agentic mesh architecture designed to foster user trust and simplify system management. LC-Opt democratizes access to detailed, customizable liquid cooling models, enabling the ML community, operators, and vendors to develop sustainable data center liquid cooling control solutions.

## 1 Introduction

The rapid growth of data-intensive applications in artificial intelligence (AI), high-performance computing (HPC), and cloud services has driven a sharp rise in data center energy demand. With increasing server power density, traditional air cooling has become both thermally and economically inadequate [1], particularly for modern CPUs [2] and GPUs [3]. This shift has accelerated the adoption of liquid cooling (LC), which offers higher heat removal efficiency and can cut cooling

---

[*]Corresponding author. [†]These authors contributed equally.

39th Conference on Neural Information Processing Systems (NeurIPS 2025) Track on Datasets and Benchmarks.

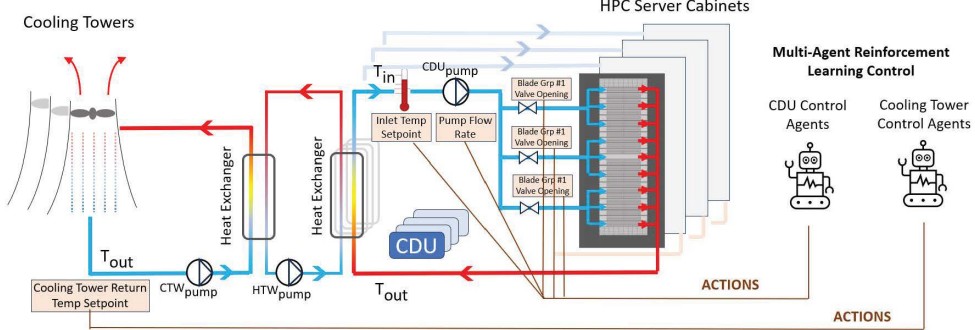

**Figure 1:** System Overview of end-to-end Control of Liquid Cooled Data Center. The CDU RL agents control the HPC server cabinets. The Cooling Towers are controlled by the CT RL agents.

energy and carbon footprint by up to $63\%$ [4]. However, LC systems largely rely on static or rule-based controls [5, 6, 7], limiting their energy-saving potential.

Optimizing LC in HPC data centers is complex due to tight coupling with HVAC subsystems—CDUs, heat exchangers (HX), cooling towers (CT), and pumps [8, 9, 10]. Designing dynamic rule-based strategies for such systems is impractical. While Deep RL has shown promise in domains like robotics and HVAC, its application to large-scale LC systems is constrained by the absence of realistic testbeds. To address this, we introduce **LC-Opt**, a benchmarking framework extending Oak Ridge National Lab's (ORNL) high-fidelity Modelica Digital Twin (DT) of the Frontier supercomputer [11] with RL-ready control interfaces. Code, licenses, and setup instructions for LC-Opt are available at GitHub[2]. The main contributions of this work are:

1. **Liquid Cooling Control Benchmark:** Evaluate control strategies for an HPC data center with ORNL's **Frontier supercomputer**'s digital twin as baseline and additional features.

2. **End-to-end Customizable and Scalable Benchmark:** Supports energy optimization of fine-grained Blade-Group (BG) server temperature control, cooling tower (CT), and heat reuse (HRU).

3. **Gymnasium interface:** Supports RL, LLM-based, and traditional controllers (ASHRAE G36 [12]) for real-time control strategies.

4. **Multi-agent Control:** Supports single-agent, multi-agent, homogeneous, and hybrid RL policies in customizable data center setups.

5. **Granular & Hybrid Control:** CDU control agents regulate HPC cabinet temperatures by adjusting **inlet temperature setpoints, pump flow rates, and blade-group valve openings**. Cooling Tower control agents minimize energy consumption by modulating **return water temperatures**.

6. **Ablation and scalability:** Reference evaluation of agents and scalability for large-scale cooling systems, and hybrid policy effectiveness for blade-level cooling control and cooling tower.

7. **Policy interpretability:** Supports **model distillation** [13] for policy extraction with LLM and decision tree-based strategies, aiding researchers and operators with policy validation.

8. **LLM explainability and Agentic Design:** Supports **LLM controllers** that generate natural language explanations of control actions to enhance operator trust and simplify system management.

## 2   Related Work

The growing energy demands of data centers and their increasing impact on global carbon emissions have driven research to improve cooling systems. Traditional cooling methods, such as static setpoints and basic rule-based controllers [14, 15], have been widely used in the past. Although these methods helped ensure stable temperatures, they were not efficient enough to handle modern data centers' complex and changing conditions. Liquid cooling (LC) has become an efficient alternative to air cooling due to its ability to transfer heat more efficiently and handle larger workloads [16, 17]. However, managing LC systems is challenging due to the need to control many factors, such as flow

---

[2]GitHub repository: https://github.com/HewlettPackard/sustain-lc.

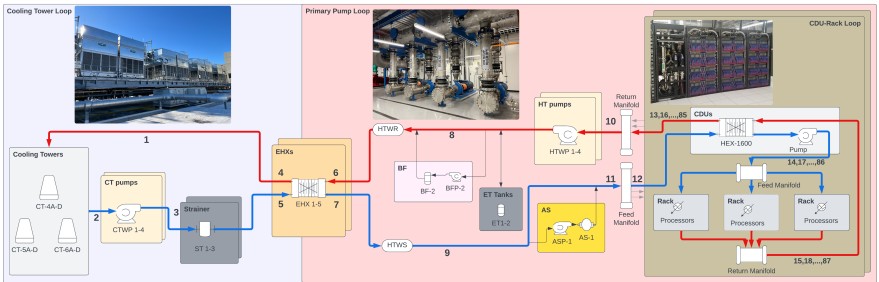

**Figure 2:** *Frontier's* Cooling System [11]

rates, temperatures, and valve settings [18]. Traditional methods often fail to fully optimize these systems, especially when workloads exhibit severely skewed allocation across clusters. Machine learning (ML) methods have been increasingly applied for predictive thermal management [19, 20] and control [21, 22], using historical and current data to perform fine-tuned cooling. While RL has been effective in optimizing air-cooled data centers, its application in LC systems is still emerging. [11] presents a framework demonstrating modeling of transient thermo-fluidic dynamics and energy efficiency for liquid-cooled exascale supercomputers, in Modelica. This work significantly extends the Modelica model with a fine-grained AI enabled control interfaces wrapping it in a Gymnasium-compatible framework, enabling scalable RL and generative AI-based control for high-fidelity, energy-efficient cooling—advancing beyond static rules like ASHRAE Guideline 36 [12].

## 3   LC-Opt Description

First, in Section 3.1, we provide a high-level description of the Frontier Liquid Cooling system and how we have augmented the compute block with blade-level cooling, coolant setpoint and flow rate control, and Cooling Tower water setpoint control. This enables machine learning-based temperature and power management. Then we provide the implementation details of the modeling (Section 3.2) and control problems (Section 3.3) included in LC-Opt.

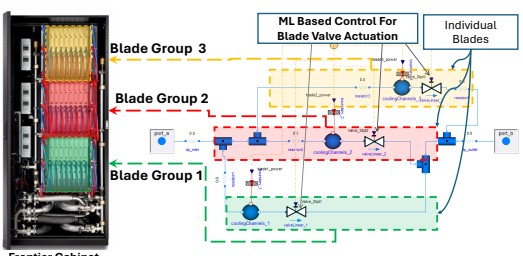

**Figure 3:** Blade Level Cooling control is one of the Modelica modeling augmentations to the Frontier Liquid Cooling system model.

### 3.1   Frontier Liquid Cooling System for Machine Learning Applications

Figure 2 illustrates the Frontier Liquid Cooling (LC) system. On the right, there are arrays of cabinets that house servers (in racks), grouped into blades. Each cabinet is paired with a Cooling Distribution Unit (CDU-Rack Loop), which extracts heat via a heat exchanger(HX) and pumps by adjusting coolant temperature and flow rate. Frontier includes 25 such CDU-Rack pairs, transferring heat to a central Hot Water System (HTW, center of figure). This heat is ultimately rejected to the environment via the Cooling Tower (CT) Loop, which employs fans for forced-draft cooling through sensible and latent processes (Cooling Tower Loop in the figure). CT power consumption depends on wetbulb temperature, supply temperature setpoint, and incoming thermal load. The system also features a Heat Recovery Unit (HRU), an intermediate heat exchanger that reuses server waste heat for ancillary heating (e.g., residential or district applications).

The original Modelica model employed ASHRAE-based or static rule-based controllers [12]. We augmented the model to support RL-based control to enhance temperature regulation at CDUs and optimize energy use at CTs. The most elaborate augmentation is the blade group (BG) control for each cabinet, as shown in Figure 3. A blade group is a collection of servers that are served by one branch of the cooling liquid. For the Cooling Tower, traditional staging and return temperature controls were replaced with RL-based supervisory control(Cooling Tower ML Control). Similarly, for CDU-cabinet pairs, RL agents now regulate coolant temperature setpoints, flow rates (CDU ML Controls). These two augmentations are further shown in Figure 11 in Appendix C for advanced Modelica users.

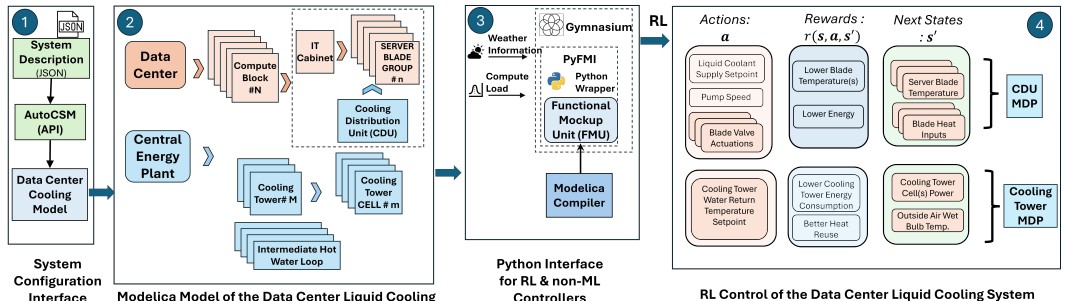

**Figure 4:** Functional Design of LC-Opt. **1) AutoCSM API to create a Modelica model** for a Data Center Cooling system based on the System Description in JSON format. **2) Hierarchical structure of the cooling system model**. **3) LC-Opt *Gymnasium* environment** provides the Python interface for the Modelica Functional Mockup Unit (FMU) binary executable. **4) MDPs for multi-agent RL** cooling control.

## 3.2  LC-Opt Modeling

The model implementations in LC-Opt are based on the liquid cooling (LC) system description provided in Section 3.1. Parts 1, 2, and 3 in figure 4 comprise the modeling process.

1. We first specify the composition of the LC system in a JSON file in a hierarchical manner. This allows us to build customizable LC setups that include individual cabinets, the cooling distribution units (CDU), the heat exchangers for intermediate as well as heat reuse primary and secondary loops, valves, pumps, sensors, and finally, the cooling tower. It further allows us to specify components that define the nature of control, whether rule-based or RL-enabled, the data sources to be used as exogenous variables, the system parameterization, etc. The components of the LC system specified in the JSON file are then read using the *AutoCSM* API [23], and it creates a Modelica model of the data center liquid cooling system

2. The system's hierarchical specification enables the creation of complex yet interpretable LC architectures. Advanced users can visualize and modify system details via Modelica IDEs (e.g., Dymola, OpenModelica), using components from the *datacenterCoolingModel* library [10]. LC-Opt uniquely supports blade-group (BG) valve control, allowing fine-grained actuation based on heat input to the blade groups. To our knowledge, this is the first framework offering detailed modeling and benchmarking of hybrid control, serving both ML researchers (e.g., multihead policies) and data center practitioners.

3. To enable ML-based control of the LC system, we exported the Modelica model to a Functional Mockup Unit (FMU) using a Modelica compiler. The FMU, a non-linear state-space system, integrates with Python frameworks like Gymnasium [24]. It accepts both exogenous and control inputs, updates its internal state, and produces outputs for downstream use. Python interfaces to the FMU are standardized via FMPy [25] and PyFMI [26]. Exogenous variables simulate non-control dynamics such as heat generation at blade groups or external weather, influencing system behavior like cooling tower performance. It supports various FMUs to test custom environments and evaluate RL scalability, particularly for hybrid action spaces in HPC. A detailed guide for compiling custom LC-Opt environments is included in the Supplemental and GitHub Readme.

## 3.3  LC-Opt Control

4. To facilitate control applications for the LC system, LC-Opt wraps the FMU with the Gymasium interface. We primarily focus on two problems for LC-Opt. Firstly, at the higher level, we wish to reduce the overall data center energy cooling consumption, which is dominated by the CTs. Hence, we build a CT Markov Decision Process (MDP) as shown in Figure 4. The detailed formulation of the MDP is shown further in Table 1. The other goal is to ensure optimum operating temperatures of the blade groups. Hence, we create the Blade Group Level MDP that focuses on this aspect of the problem. Details of the corresponding MDP are provided in Table 2.

A key aspect of LC-Opt is that both MDPs share the same FMU transition model ($\mathcal{T}$), enabling mutual influence. However, due to weak thermal coupling (Figure 2) and sheer state space size, single-agent or centralized training with decentralized execution (CTDE) for multiagents was ineffective and

**Table 1:** Cooling Tower MDP

| MDP Attributes | Formulation | Remarks |
|---|---|---|
| State ($s_t$) | $\begin{bmatrix} P_{11} & \dots & P_{1j} & \dots & P_{1t} & T_{ct,1} \\ \vdots & & \vdots & & \vdots & \vdots \\ P_{i1} & \dots & P_{ij} & \dots & P_{it} & T_{ct,i} \\ \vdots & & \vdots & & \vdots & \vdots \\ P_{N1} & \dots & P_{Nj} & \dots & P_{Nt} & T_{ct,N} \quad T_{wb} \end{bmatrix}$ | $P_{ij}$ refers to the power consumption of the $j^{th}$ cell of the $i^{th}$ cooling tower. $T_{ct,i}$ refers to the $i^{th}$ cooling tower water return temperature. $T_{wb}$ is the outside air wet bulb temperature |
| Action ($a_t$) | $\delta_1, \dots, \delta_i, \dots, \delta_N$ | The agents sets the changes in cooling tower water return temperature setpoint $T_{ct,i}$ by $\delta_i$ across all the N cooling towers |
| Reward ($r_t(s_t, a_t, s_{t+1})$) | $-\sum_{i,j} P_{i,j}$ | It is the sum total of the power consumption across all the cells for all the cooling towers |

**Table 2:** Blade Group Level MDP

| MDP Attributes | Formulation | Remarks |
|---|---|---|
| State ($s_t$) | $\begin{bmatrix} T_{11} & \dots & T_{1j} & \dots & T_{1B} & P_{11} & \dots & P_{1j} & \dots & P_{1B} \\ T_{i1} & \dots & T_{ij} & \dots & T_{iB} & P_{i1} & \dots & P_{ij} & \dots & P_{iB} \\ T_{C1} & \dots & T_{Cj} & \dots & T_{CB} & P_{C1} & \dots & P_{Cj} & \dots & P_{CB} \end{bmatrix}$ | $T_{ij}$ and $P_{ij}$ refer to the temperature and thermal power input respectively of the $j^{th}$ blade group of the $i^{th}$ cabinet. |
| Action ($a_t$) | $\begin{bmatrix} v_{11} & \dots & v_{1j} & \dots & v_{1B} & T_{cdu,1} & Q_{cdu,1} \\ v_{i1} & \dots & v_{ij} & \dots & v_{iB} & T_{cdu,i} & Q_{cdu,i} \\ v_{C1} & \dots & v_{Cj} & \dots & v_{CB} & T_{cdu,C} & Q_{cdu,C} \end{bmatrix}$ | $T_{cdu,i}$ and $Q_{cdu,i}$ refer to the liquid coolant supply temperature setpoint and pump flow rate of the $i^{th}$ cabinet. $v_{ij}$ refers to the valve actuation of the $j^{th}$ blade group of the $i^{th}$ cabinet |
| Reward ($r_t(s_t, a_t, s_{t+1})$) | $-\sum_{ij} T_{ij}$ | It is the aggregate of the blade group operation temperatures |

showed poor critic convergence. We instead adopt fully independent agents for the CT and BG MDPs. This is due to the large state space and Frontier's scale ($\sim 10^4$ blade groups). To address this, we test a centralized inference per MDP during rollout, as detailed later.

# 4  RL applications on LC-Opt

## 4.1  Centralized Action Execution in Multiagent RL

As discussed in Section 3.3, the environment supports diverse RL formulations, including multi-agent strategies, where they independently control the Cooling Tower (CT) and Blade Group (BG) MDPs (Tables 1 and 2). However, standard multi-agent setups face scalability issues as the number of CTs and BGs increases, due to growing state-action spaces. To mitigate this, we implement a centralized action (CA) approach with state-action decomposition, leveraging behavioral similarity across CTs and BGs. This approach is detailed in Figure 5 for CDU and Blade Group CA RL and in Figure 12 in the Appendix. for Cooling Tower CA RL. The BG MDP observation is decomposed per blade group. While BGs within a cabinet are interdependent due to fan dynamics and workload allocation, BGs across cabinets are conditionally independent [27]. For the CT MDP, we partition the observation space per tower, enabling batch inference across towers. Since CT operations are loosely coupled, primarily through shared power consumption, we treat CT power as a blocking d-separation (directional separation) [27] variable. However, return water temperature remains partially coupled across towers. To account for this, we include all return temperatures in each decomposed observation. Wet bulb temperature is also retained to capture humidity constraints on cooling capacity. For each MDP, batched inference enables parallel rollout and value estimation. To prevent experience contamination across parallel batches, we allocate separate rollout buffers per CT and cabinet, avoiding data overlap. While this batching improves inference efficiency, it introduces a limitation in terms of the memory cost, proportional to the data center's scale.

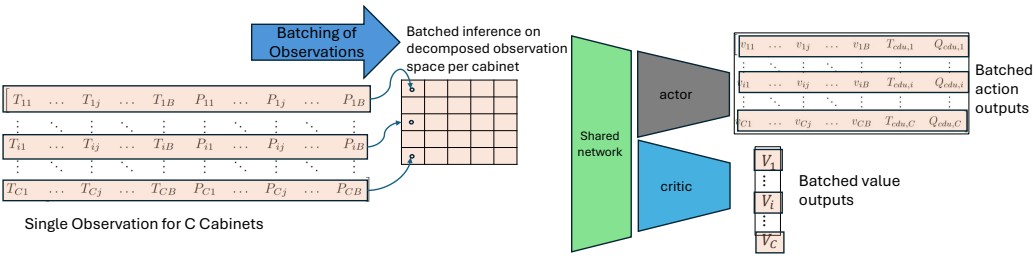

**Figure 5:** Centralized Action Execution Approach for scalable inference and rollouts at the CDU(s) and Blade Group(s) for HPC scale data center Digital Twins in LC-Opt

## 4.2  Improved Reward feedback via Multi-Head Policy

Under an ideal scenario, the valve opening for each blade group will actuate in proportion to its heat input. Discovering such a heuristic for RL using a scalar reward as in Table 2 may be extremely difficult by just looking at temperature feedback. Moreover, the agent also needs to generate a liquid coolant supply temperature setpoint for the heat exchanger and the pump flow rate. Lastly, the valve actuations have to be between $[0, 1]$ and sum up to 1.0 for conservation of mass for the FMU simulation step. Hence, we create an RL variant with a multiheaded policy for the Blade Group MDP where the first head generates the scaled temperature setpoint and flow rate in the $[-1, 1]$ range using tanh activation. The second head has a softplus output fitted to a Dirichlet distribution to generate

the desired valve response vector that is [0,1] scaled and sums to 1.0. This formulation of the actor network empirically allows for better interpretation of the reward feedback per head.

# 5 LLM-LC: An Agentic-AI Platform for Real Time Operation of Liquid Cooled Data Centers

To address the complex, multi-objective challenge of optimizing liquid cooling in High-Performance Computing (HPC) data centers, we introduce an Agentic LLM-based Digital Twin architecture. Interpretable and explainable LLM controller fine-tuned using distillation from trained RL policies serve as the basis for the framework. It moves beyond monolithic, "black-box" controllers by decomposing the control problem into a collaborative ecosystem of specialized, LLM-powered agents.

## 5.1 Policy Distillation of the Trained RL Agents in to LLMs and Decision Trees for explainability

Deep Reinforcement Learning (DRL) agents have shown strong performance in sequential decision-making, but their lack of interpretability limits deployment in safety-critical domains such as data centers. We hence show, how LLMs can be fine-tuned to perform interpretable control and generate explainable policies. This is primarily achieved by distilling where trained RL policy Experience Data is used to fine-tune the performance of an Instruction-tuned LLM as shown in Figure 6. [28]

We employ Parameter-Efficient Fine-Tuning (PEFT), specifically the QLoRA method, to efficiently adapt the LLM to mimic the oracle's decision-making process [29]. The resulting fine-tuned LLM not only functions as a controller but can also provide natural language explanations for its actions, bridging the gap between the high performance of RL policies and the need for transparency in critical applications. As an alternative, Decision Trees (DTs) also offer interpretable policies and are used to distill RL policies as part of ablation in our results. The process to generate the Experience Data in Figure 6 is discussed in details in the Appendix A

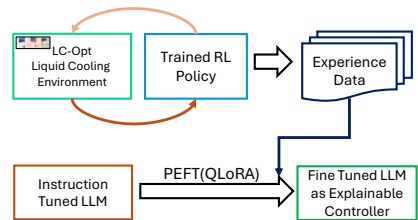

**Figure 6:** Policy Distillation in to LLMs from trained RL policy

## 5.2 Multi-Agent System for Explainable Control

As illustrated in Figure 7, the agents communicate over a central message bus to achieve a control system that is not only high-performing but also inherently transparent, resilient, and explainable. Our architecture is composed of several distinct classes of agents and tools, each responsible for a specific aspect of the control and monitoring process.

- **Reasoning & Decision Agents:** At the heart of the system is the **Control Agent**, a reasoning engine responsible for generating real-time control actions. This agent's policy is fine-tuned using deep Reinforcement Learning (RL) on a high-fidelity digital twin of the cooling system. It processes system states and leverages an LLM guided control model to make decisions. It is complemented by a **Sensor Agent**, which serves as the system's interface to the digital twin, monitoring system states, performing necessary data preprocessing, and transmitting the Control Agent's actions to the actuators.
- **Maintenance Agents:** To ensure long-term robustness and system health, a suite of maintenance agents operates continuously. The **Agent Monitor** performs meta-level oversight, tracking the performance and resource utilization of the other agents and recommending optimization strategies. The **Maintenance Agent** focuses on the physical system's integrity, monitoring health metrics, identifying anomalies and trends in thermal dynamics, predicting potential failures, and prescribing preventative maintenance actions.
- **Planning & Interface Agents:** High-level strategic management is handled by the **Configuration Agent**, which manages system configurations such as scaling, control parameters, and the hyper-parameters of the LLM agents. Crucially for human-in-the-loop interaction and trust, the **User Interface Agent** provides deep system transparency. Its **Visualization** component offers natural

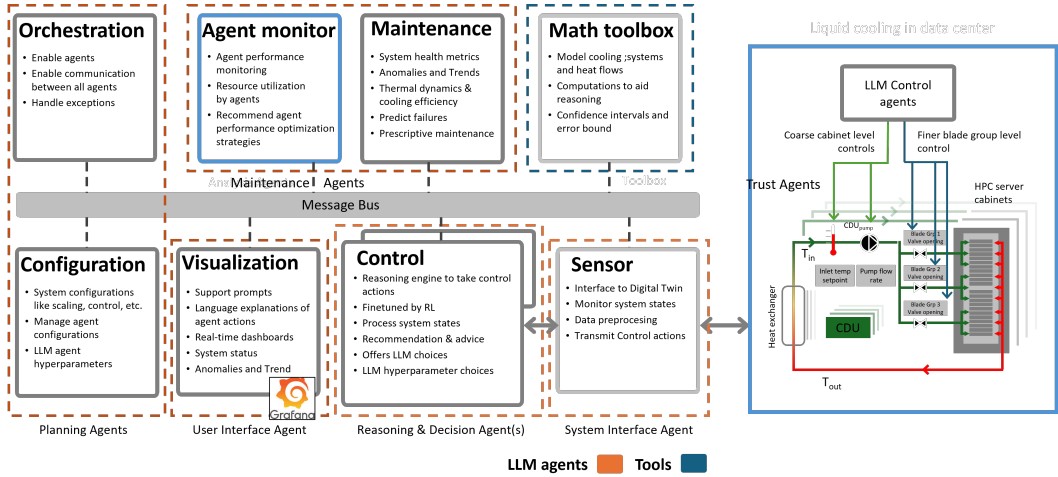

**Figure 7:** Agentic LLM based Digital Twin for Liquid Cooling explaining actions

language explanations for agent actions, presents real-time dashboards (e.g., via Grafana), and surfaces system status and trends, making the entire decision-making process intelligible to human operators.

- **Trust Agents and Tools:** The entire system is coordinated by an **Orchestration Agent**, which enables and manages inter-agent communication and handles exceptions, ensuring the collective functions as a cohesive unit. To ground the LLM's reasoning in physical reality, the agents have access to a **Math Toolbox**. This tool provides validated models of the cooling systems and heat flows, allowing agents to perform computations that aid reasoning and generate confidence intervals, ensuring that all decisions are both physically plausible and safe.

## 6  Evaluation Metrics, Hyperparameters and Experimental Settings

Based on the reward functions in Tables 1 and 2, we evaluate performance using two main metrics. First, the aggregate power consumption at the cooling tower, $\sum_{i,j} P_{i,j}$, where $P_{i,j}$ is the power used by the $j^{th}$ blade group under the $i^{th}$ cooling tower, reflects energy efficiency—particularly under high workload conditions. Second, blade-group thermal performance is measured by $D_{\text{blade},t} = 100 \times \frac{\mathcal{I}[(\mathcal{U}_T > \bar{T}_t) \wedge (\bar{T}_t > \mathcal{L}T)]}{||\bar{T}_t||}$, the percentage of time blade temperatures $\bar{T}_t = [T_{i,j}]$ remain within nominal bounds $\mathcal{L}_T$ and $\mathcal{U}_T$, based on manufacturer specifications [30]. The lower bound avoids overestimating performance during idle periods. For reference, we also compute the total cooling energy per CDU via $Q_i$ for cabinet $i$. The tuned hyperparameter details are provided in the Appendix Tables 9 for the multihead (hybrid actions) blade group control policy and Table 10 for the discrete control policy of the cooling tower. Experiments were run on an Intel® Xeon® Platinum 8470 server (104 CPUs, 1×H100 GPU), with 12 threads per training agent.

## 7  Results

### 7.1  Ablations on agents, and scaling

We evaluate the control strategies (as described in Sections 4.1 and 4.2) using the metrics defined in Section 6. Table 3 summarizes these strategies along with their performance. A radar chart representation of the relative performance is shown in Figure 8 for better visualization. All RL agents were trained using Proximal (PPO) [31] while for the Baseline, we developed a "trim and respond" logic based on the industry standard ASHRAE Guideline 36 [12]. The baseline control logic developed for Liquid Cooling and Cooling Towers is described in detail in the Appendix K. We also provide the **cumulative carbon footprint** of these approaches over a 2-day period in Table 16 in the Appendix

**Ablation:** For the experiments performed, Cases 1–4 demonstrate that RL without valve-level control (Case 3) underperforms in maintaining the desired temperature range compared to the G36 baseline (Case 1). Moreover, incorporating RL at the CT level while retaining G36 at the BG level (Case 2) increases CT power consumption due to aggressive valve actions by the G36 policy. Cases 5–7 highlight the benefits of multi-agent control: centralized actions (Case 5), state-space and action centralization (Case 6), and multi-head policy architecture (Case 7) progressively improve temperature regulation while reducing CT power usage. These results suggest that enhanced control at the BG level cascades to better overall cooling performance.

**Table 3: Ablation of RL Agent Design with PPO.** We incrementally replace the static baseline (Case 1) with RL controllers for: Cooling Tower (Case 2), CDU coolant setpoint/flow (Case 3), and Blade Group valves (Case 4). Case 5 introduces a single multi-agent RL controller, Case 6 adds batching for state space reduction, and Case 7 uses a multi-head policy. Experiments use $N$=2 towers, $m$=2 cells, $C$=5 cabinets with $B$=3 blade groups each, and are evaluated on an unseen exogenous trace. Blade-group temperature compliance $D_{blade,avg}$ is computed with $\mathcal{U}_T$=40°C and $\mathcal{L}_T$=20°C.

| Metric → 

 Agent/Control Type ↓ | Control Details | $D_{blade,avg}\%$ 
 (% of time Temp within ideal range) | $\sum P_{ij}(kW)$ 
 (Cooling Tower Avg Power) | $\sum Q_i$ 
 (IT Level Avg Cooling Power) | **Avg Episode Reward** 
 **per** 
 **Cabinet** | **per** 
 **Cooling** 
 **Tower** |
|---|---|---|---|---|---|---|
| 1. Baseline Control | ASHRAE G36 | 76.92 | 237.31 | 235.28 | 1697.08 | 360.17 |
| 2. CT RL + BG Baseline | Only CT RL control | 79.21 | 246.46 | 235.03 | 1702.16 | 352.28 |
| 3. CT Baseline + BG RL | No Valve Control | 64.91 | 217.6 | 203.96 | 1638.48 | 372.97 |
| 4. CT Baseline + BG RL | With Valve Control | 77.13 | 217.37 | 211.83 | 1698.36 | 373.52 |
| 5. Multiagent RL | Decentralized Action | 78.24 | 218.11 | 212.94 | 1697.49 | 370.51 |
| 6. Multiagent RL | Centralized Action (CA) | 90.46 | 207.37 | 208.69 | 1714.65 | 395.88 |
| 7. Multiagent RL | CA & Multihead policy | **95.63** | **206.52** | **197.18** | **1726.31** | **396.24** |

**Table 4: Performance on Scale.** Evaluation of Rule-Based Control vs Multihead Centralized Action Policy for Scaling of Cooling Tower Agent and Multi-head Blade-Group Agent with increasing Data Center sizes. Blade-Group Agent is trained on N=2 Cooling Towers, m=2 Cells per Tower, C=5 Cabinets, B=3 Blade Groups per Cabinet

| Metric | | N=2, m=2 C=10, B=3 | N=2, m=2 C=15, B=3 |
|---|---|---|---|
| $D_{\text{blade,avg}}$ % | ASHRAE G36 | 71.92 | 68.24 |
| | CA & MH Policy | **96.28** | **86.19** |
| $\sum P_{ij}$ (kW) | ASHRAE G36 | 390.26 | 548.61 |
| | CA & MH Policy | **388.84** | 560.72 |
| Metric | | N=3, m=2 C=20, B=3 | N=4, m=2 C=25, B=3 |
| $D_{\text{blade,avg}}$ % | ASHRAE G36 | 75.31 | 83.08 |
| | CA & MH Policy | **94.07** | **92.61** |
| $\sum P_{ij}$ (kW) | ASHRAE G36 | 922.7 | 1381.92 |
| | CA & MH Policy | **871.26** | **1109.84** |

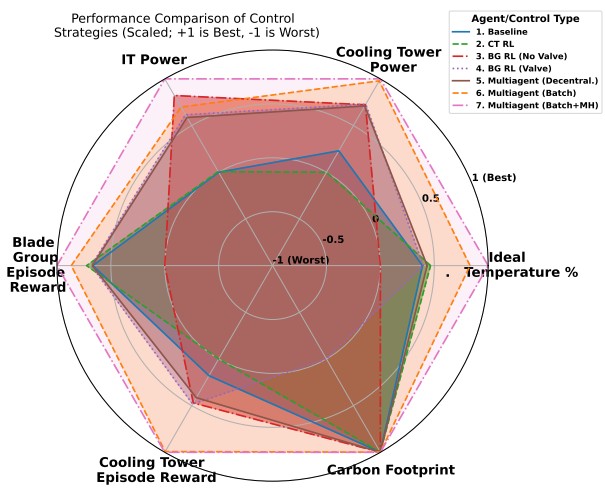

**Figure 8:** Relative Performance of different RL approaches from Table 3 for $N = 2$ towers, $m = 2$ cells, $C = 5$ cabinets, with $B = 3$ blade groups in each cabinet

**Scalability:** Table 4 compares the performance of the ASHRAE G36 rule-based controller versus the Centralized Action (CA) Multihead RL policy on unseen larger liquid cooling configurations on which it was never trained. The RL policy consistently maintains blade group temperatures within the ideal range defined by $\mathcal{U}_T$ and $\mathcal{L}_T$, while Guideline 36 shows variability in $D_{blade,avg}$ due to inconsistent cooling tower provisioning. Analysis of evaluation timeseries indicates that RL agents effectively allocate cooling based on blade group heat load, whereas the baseline wastes cooling on idle or low-load groups, causing temperature violations in both under- and overprovisioned regions.

## 7.2 Multihead Policy: Discovery of Optimal Blade Control

The multi-head policy enables the agent to align valve actuation with blade group power levels, dynamically allocating more coolant to blades with higher power inputs. This adaptive behavior, particularly evident in Blade Groups 2 and 3 (as shown in Figure 9), enhances cooling efficiency. The relationship discovered by the agent, between heat inputs and RL valve actuations, appears non-linear, consistent with the quasi-periodic nature of power input, as reflected in both linear and non-linear correlation metrics in Table 5. However, Figure 9 also shows reduced valve actuation for Blade Group 1, correlating with its lower power profile, occasionally resulting in slightly elevated temperatures for Blade Group 1. It is a limitation we aim to address in future work.

**Table 5: Relation between Power Input and RL Valve Actuations**. Correlation Coefficients

| Agent Type | Correlation Metric | Value |
|---|---|---|
| | Pearson Coefficient ($\rho$) | 0.051 |
| Single head | Spearman Rank ($\rho_s$) | 0.215 |
| Blade Group Agent | Mutual Information (MI) | 1.831 |
| | Pearson Coefficient ($\rho$) | 0.136 |
| Multi-head | Spearman Rank ($\rho_s$) | 0.680 |
| Blade Group Agent | Mutual Information (MI) | 2.653 |

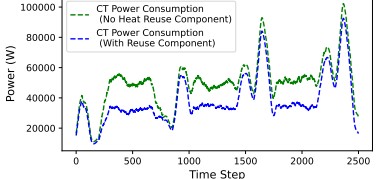

**Table 6:** Difference in Centralized Action Multihead RL (Case 7) Energy Consumption of the Cooling Tower without and with Heat Recovery block

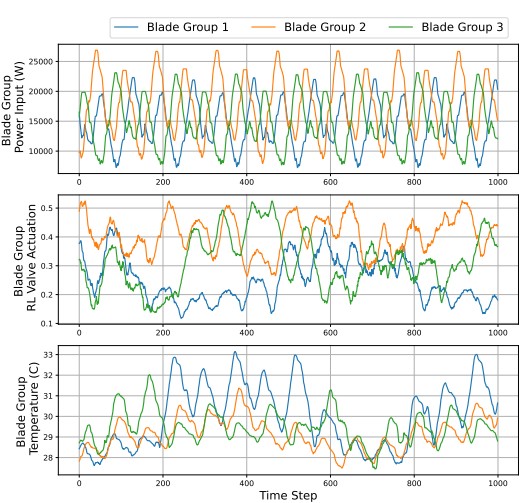

**Figure 9:** Snippet of Power Input, Multihead policy Valve Actuation and Blade Group Temperature

## 7.3 Heat Recovery Unit for Lower Cooling Tower Power Consumption

The benchmark provides an option for the addition of the heat recovery unit (HRU) to evaluate how the power consumption in the cooling tower can be reduced. The multihead policy (Case 7) has been evaluated on LC-Opt models with and without the heat recovery models discussed in section 3.2. The addition of HRUs directly reduces average power consumption by approximately $21\%$ (10.2kW average evaluated over 17 hrs) as shown in Figure 6 for a medium-sized data center(N=2, m=2, C=5, B=3), and the recovered heat can be used for ancillary functions like commercial and residential spaces or district heating applications. In a future work, we wish to include models to simulate the effects and efficiency of this heat ducted away by HRUs on such spaces.

## 7.4 Performance comparison of finte-tuned LLMs from trained RL agents

We distilled pretrained policies (Section 5.1) into LLMs and decision trees for the Cooling Tower (CT, discrete actions) and Blade Groups (BG, continuous actions). We perform an ablation with a rule-based baseline (ASHRAE), trained PPO (RL), and variations of fine-tuned LLMs distilled from the RL agent in Figure 10. Fine-tuning the LLMs on the RL agent's trajectories, particularly the Qwen model, resulted in the best overall performance, achieving the lowest average blade temperatures and cooling tower power consumption. These distilled LLM controllers not only significantly surpassed the ASHRAE baseline and the base LLMs with few-shot prompting but also outperformed the original RL policy, demonstrating the effectiveness of this knowledge transfer. Similar analysis for Decision Trees is shown in Appendix J

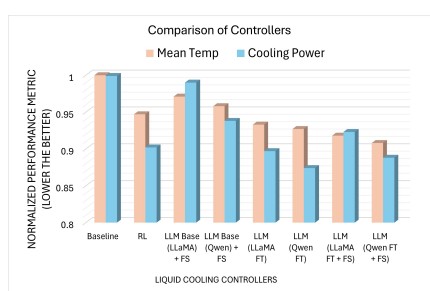

**Figure 10:** Ablation of Distilled LLM policies

### 7.5 LLM Explainability and Validity of Explanation

We adopt a human-as-a-judge evaluation approach, where domain-aware annotators assess the explanation based on two criteria: (i) faithfulness — whether the explanation aligns with the given observation and the RL action (ii) completeness — whether the explanation accounts for all relevant dimensions of the action (iii) if the generated explanation is theoretically consistent and factually correct.

---

**LLM Explainability Example**

Obs (315.45, 314.04, 311.25, 46806.57, 46806.57, 31632.4), Act: (0.24,0.35,0.41,40.95,24.66)
**Explanation**: Given that thermal readings remain within tolerable bounds, the agent increases the setpoint to 40.95 for energy conservation. Coolant flow is distributed with branch 3 receiving the most, targeting demand while sidestepping unnecessary cooling. This approach supports environmental compliance standards while ensuring uptime. Reducing cooling overheads has cascading social benefits, especially in energy-constrained regions.
**Validity**: 1) Human Expert Evaluation: "While the LLM response correctly attributes the increased temperature setpoint of the coolant due to moderate temperatures in the cabinet, it does not completely explain the other values that were generated by the reinforcement learning agent. Also, it does not explain why the current distribution of the fluid happens across the three branches"

---

**LLM Explainability and Guidance Example**

Obs: (307.2, 306.74, 309.77, 21567.55, 44779.22, 33672.41), Act: (0.19, 0.61, 0.2, 22.39, 25.52)
**Explanation**: Branch 2 remains relatively cool, allowing the controller to redistribute cooling capacity toward more loaded or hotter branches. A setpoint of 22.39 and a controlled flow of 25.52 signify this adaptive behavior. The decision not only maximizes performance-per-watt but also aligns with global decarbonization efforts. It is recommended to reduce branch 1 coolant allocation to 0.1 and to allocate the rest to branch 3 since it has a higher workload
**Validity**: 1) Human Expert Evaluation: "This is one of the better examples where the agent is explaining the actions for all three blade groups. It also, provides a good way of reifining the RL valve actions"

---

## 8 Conclusion

We present a configurable benchmark for end-to-end liquid-cooled HPC data centers, extending the Frontier supercomputer's baseline model. The framework integrates a detailed Modelica-based control interface with a Gymnasium-compatible RL environment, supporting both RL and traditional controllers with fine-grained actuation. It features a formal MDP setup, scalable multi-agent RL implementations, heat recovery support, and tools for model distillation and LLM-based explanation of control actions. Building on this foundation, we introduce **LLM controllers** as explainable agents that articulate control decisions in natural language, extending the technology frontier through an *agentic design* that coordinates multiple LLMs for comprehensive and user-friendly system management. This accessible platform empowers ML researchers and practitioners to advance sustainable, energy-efficient control strategies.

The primary goal of this paper is to establish a sustainability benchmark grounded in a high-fidelity, real-world data center model—exemplified by the Frontier supercomputer digital twin from the ExaDigiT consortium. Our design emphasizes generalizability through two key capabilities: (1) flexible adaptation to diverse weather and workload profiles across global locations, and (2) an open, modular architecture that allows users to customize data center configurations for benchmarking, design guidance, diagnostics, and digital twin research.

**Impact Statement** The escalating power footprint and density of successive GPU generations pose a critical challenge for AI infrastructure. This work, building on the Frontier supercomputer and advancing to server blade group-level RL control, pushes the boundaries of how reinforcement learning can address the growing cooling demands of next-generation AI systems.

**Limitations** The benchmark has a few limitations we aim to address: incorporating chip-level thermal modeling to support next-gen GPUs in high-density servers, adding hybrid cooling (air + liquid) used in some data centers, and expanding evaluation to include different weather conditions and AI workload patterns.

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

# Appendix

## Contents

# A  Experience Data Generation for LLM policy distillation

The Experience Data in Figure 6, is generated by the VIPER algorithm [13] to create interpretable and verifiable actions. The data generation process adapted to PPO is straightforward. Given that our oracle is a PPO-trained policy, we adapt VIPER to compute sample importance weights $l(s)$ consistent with PPO's stochastic policy output. Specifically, for each rollout, we record state-action pairs $(s_t, a_t = \pi^*(s_t))$ and compute $l(s) = \log \pi^*(a^*|s) - \min_a \log \pi^*(a|s)$. For Gaussian policies, the minimum log-probability occurs at the action-space boundary. We perform multiple rollouts of the deterministic oracle policy, collecting trajectories into a buffer $D$. We then resample $D$ according to $l(s)$ to obtain a weighted dataset $D'$, on which we fine tune the LLMs and separately train decision and regression trees.

# B  Ablation with Soft Actor Critic

**Table 7: Ablation of RL Agent Design with SAC.** We incrementally replace the static baseline (Case 1) with RL controllers for: Cooling Tower (Case 2), CDU coolant setpoint/flow (Case 3), and Blade Group valves (Case 4). Case 5 introduces a single multi-agent RL controller, Case 6 adds batching for state space reduction, and Case 7 uses a multi-head policy. Experiments use $N=2$ towers, $m=2$ cells, $C=5$ cabinets with $B=3$ blade groups each, and are evaluated on an unseen exogenous trace. Blade-group temperature compliance $D_{blade,avg}$ is computed with $\mathcal{U}_T=40°C$ and $\mathcal{L}_T=20°C$.

| Metric → | | $D_{blade,avg}\%$ | $\sum P_{ij}(kW)$ | $\sum Q_i$ | **Avg Episode Reward** | |
| --- | --- | --- | --- | --- | --- | --- |
| **Agent/Control Type ↓** | **Control Details** | (% of time Temp within ideal range) | (Cooling Tower Avg Power) | (IT Level Avg Cooling Power) | **per** Cabinet | **per** Cooling Tower |
| 1. Baseline Control | ASHRAE G36 | 76.92 | 237.31 | 235.28 | 1697.08 | 360.17 |
| 2. CT RL + BG Baseline | Only CT RL control | 44.61 | **208.37** | **197.41** | 1386.94 | **392.81** |
| 3. CT Baseline + BG RL | No Valve Control | 55.05 | 235.16 | 212.98 | 1528.68 | 364.35 |
| 4. CT Baseline + BG RL | With Valve Control | 57.11 | 242.86 | 224.60 | 1531.94 | 358.69 |
| 5. Multiagent RL | Decentralized Action | 79.31 | 213.84 | 228.35 | 1703.19 | 386.48 |
| 6. Multiagent RL | Centralized Action (CA) | 80.18 | 216.16 | 231.80 | 1704.88 | 381.05 |
| 7. Multiagent RL | CA & Multihead policy | **84.33** | 225.32 | 229.54 | **1709.45** | 375.49 |

## C    LC-Opt Modelica Augmentations for all components

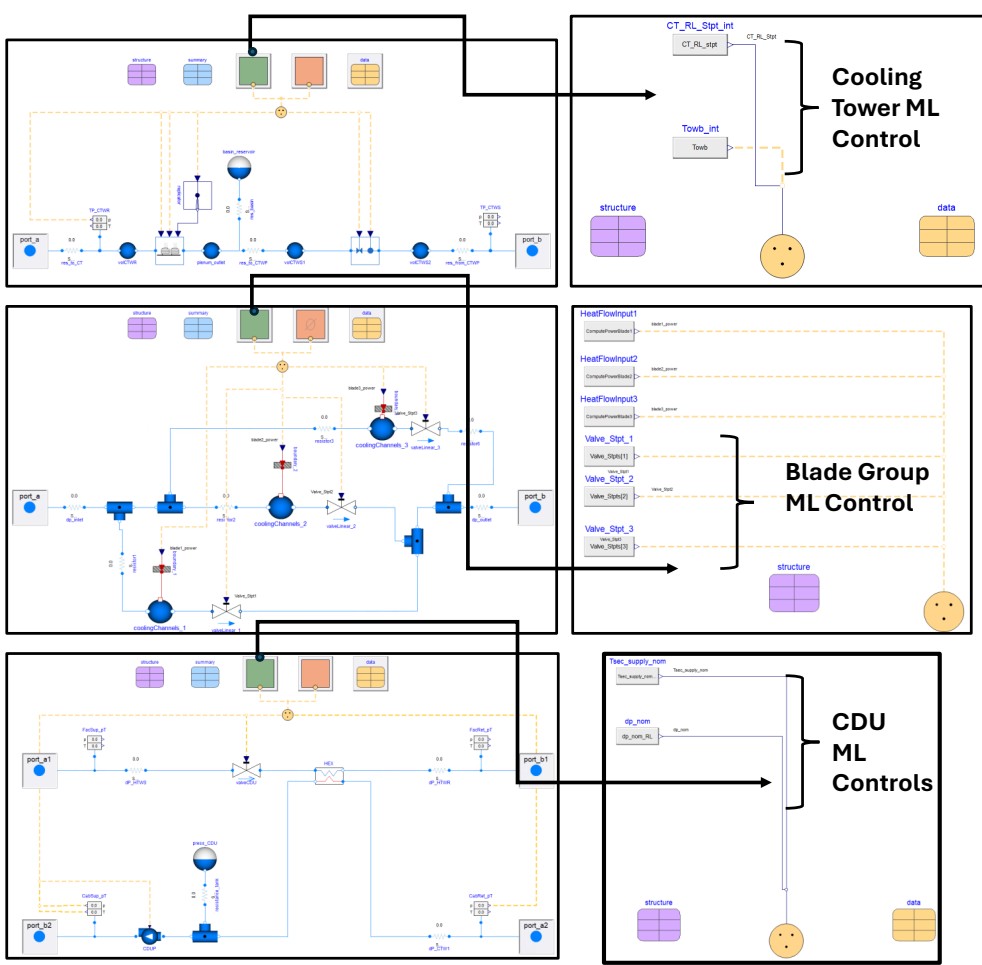

**Figure 11:** Modelica Level Augmentations to the Frontier Liquid Cooling system model to enable Machine Learning based control for the Cooling Tower, Blade Groups, and Cooling Distribution Units (CDUs)

# D   Multihead Actor Network for the Cooling Tower

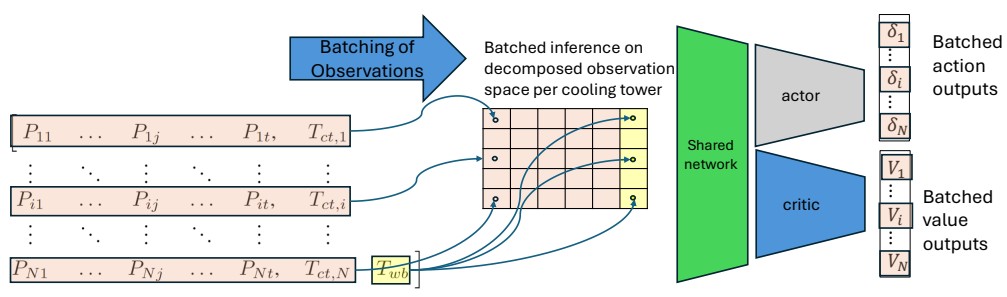

**Figure 12:** Centralized Action Execution Approach for scalable inference and rollouts at the Cooling Tower(s) for HPC scale data center Digital Twins in LC-Opt

# E  LLM for explaining agent Actions

Here we show several examples where the centralized action (CA) multihead RL policy took certain actions for the CDUs and how the agent interprets the actions. We observe that, particularly for certain cases, it is making inferences without explaining the value distributions. It at most provides explanations for the valve output, which is exhibiting the highest heat input. Particularly, the last case in the table 8 is from a timeline in figure 9 where it correctly identifies that one of the branches is starved of the cooling fluid due to low heat input.

| Multihead Policy Control Raw Action | Explanation of the Action by the LLM |
|---|---|
| Head 1 Output: (28.39°C, 27.87 kg/s); Head 2 Output (0.32, 0.35, 0.33) | Faced with consistently high temperatures—44.58°C, 41.27°C, and 42.7°C—and elevated power usage on branches 1 and 3, the system adopts a moderated approach by setting the CDU supply temperature to 28.39°C (Medium). The second branch's relatively low power likely justifies this middle-ground response. Coolant flows at 27.87 kg/s, split nearly evenly across branches (0.32, 0.35, 0.33). |
| Head 1 Output: (29.12°C, 30.06 kg/s); Head 2 Output (0.27, 0.36, 0.37) | Despite elevated temperatures across the board, the power input for branch 1 remains low. In contrast, branches 2 and 3 exhibit both high temperatures and higher power draws. The controller's response—setting a CDU supply point at 29.12°C (Medium)—appears to target the thermal load of branches 2 and 3. The flow rate of 30.06 kg/s supports this balance, with slightly more coolant sent to branch 2 (0.36). |
| Head 1 Output: (27.09°C, 27.87 kg/s); Head 2 Output (0.29, 0.36, 0.35) | Temperatures lean toward the lower end, particularly on branch 2, while power input is notably high on branch 1 alone. In light of this, the system opts for a conservative CDU temperature of 27.09°C (Low), likely sufficient to address localized heat without excessive cooling. Coolant flow is modest at 28.11kg/s, with distribution tilting slightly toward branch 3 (0.35). |
| Head 1 Output: (29.12°C, 27.82 kg/s); Head 2 Output (0.36, 0.36, 0.28) | Even though all branches exhibit Medium High thermal readings, power is concentrated only on branch 2. The system reacts by setting the CDU setpoint to 29.12°C (Medium), likely a compromise between thermal uniformity and localized electrical stress. The flow rate of 27.82 kg/s and the slightly greater coolant allocation to branch 2 (0.36) support this targeted strategy. |
| Head 1 Output: (29.42°C, 29.12 kg/s); Head 2 Output (0.26, 0.38, 0.36) | The controller observes significantly high thermal and power loads on branches 2 and 3, with only branch 1 staying relatively cooler and less active. In response, the CDU supply temperature is elevated to 29.42°C (Medium) to handle heat stress. Flow rate is maintained at 29.12 kg/s, and coolant distribution favors the high-demand branches accordingly. |
| Head 1 Output: (27.29°C, 31.17 kg/s); Head 2 Output (0.36, 0.35, 0.29) | Here, branch 2 emerges as the most thermally and electrically stressed, showing both higher temperatures and a MedHigh power draw. The system compensates with a CDU supply temperature of 27.29°C (Medium) and an elevated flow rate of 31.17 kg/s. Coolant distribution reflects this load pattern, slightly emphasizing branch 2 at 0.35. |
| Head 1 Output: (29.12°C, 27.89 kg/s); | |

| Head 2 Output (0.35, 0.38, 0.27) | Power demand is sharply skewed toward branches 1 and 2, despite moderate thermal readings. Branch 3, though cooler and less active, receives less coolant. To mitigate overheating risk on the loaded branches, the CDU setpoint is held at 29.12°C (Medium) with a flow rate of 27.89 kg/s. Distribution priorities reflect this imbalance. |
| --- | --- |

# F   Policy Hyperparameters

**Table 9:** Tuned Hyperparameters for Multi-Head Centralized Actor PPO (Used for Blade Group and CDU Controls)

| Hyperparameter | Value |
| --- | --- |
| Learning Rate (Actor) | 0.0003 |
| Learning Rate (Critic) | 0.001 |
| Discount Factor ($\gamma$) | 0.80 |
| PPO Epochs ($K$) | 50 |
| Clipping Parameter ($\epsilon$) | 0.2 |
| Initial Action Standard Deviation | 0.6 |
| Actor Hidden Layers | [64, 64] |
| Actor Activation Function | Tanh |
| Critic Hidden Layers | [64, 64] |
| Critic Activation Function | Tanh |
| Top-Level Action Head | Tanh |
| Valve-Level Action Head | Softmax + Dirichlet |
| Minibatch Size | 32 |
| Entropy Coefficient | 0.01 |
| Value Function Coefficient | 0.5 |
| Action Std Decay Rate | $5 \times 10^{-4}$ |
| Minimum Action Std | 0.1 |
| Total Time steps | $2 \times 10^{6}$ |
| Update Interval | 2048 steps |

**Table 10:** Tuned Hyperparameters for Centralized Actor PPO (Used for Cooling Tower Control)

| Hyperparameter | Value |
| --- | --- |
| Learning Rate (Actor) | 0.0006 |
| Learning Rate (Critic) | 0.001 |
| Discount Factor ($\gamma$) | 0.95 |
| PPO Epochs ($K$) | 50 |
| Clipping Parameter ($\epsilon$) | 0.2 |
| Initial Action Standard Deviation | 0.6 |
| Actor Hidden Layers | [32, 64] |
| Actor Activation Function | Softmax |
| Critic Hidden Layers | [32, 32] |
| Critic Activation Function | Tanh |
| Minibatch Size | 32 |
| Entropy Coefficient | 0.01 |
| Value Function Coefficient | 0.6 |
| Action Std Decay Rate | $5 \times 10^{-4}$ |
| Minimum Action Std | 0.1 |
| Total Time steps | $2 \times 10^{6}$ |
| Update Interval | 2048 steps |

## G    Correlation Plot for Multihead Policy Blade Group Valve control

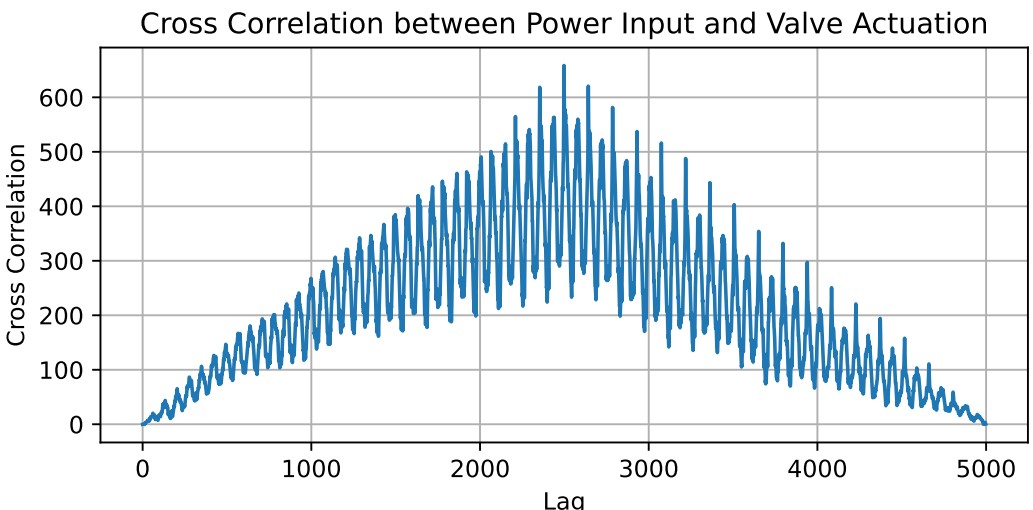

**Figure 13:** Correlation Coefficients under Multi-head Policy

## H    Bridging the Simulation-to-Real gap

The digital twin at the core of SustainLC is not a theoretical model; it is a high-fidelity simulation that has been validated against the operational dynamics of the Frontier supercomputer's cooling system. This validation provides the confidence needed for its use in developing next-generation control strategies.

The primary scope of a benchmark is to provide a standardized, accessible, and risk-free environment for developing and comparing novel control strategies—something that is infeasible and cost-prohibitive on a live, multi-million dollar production supercomputer. SustainLC provides exactly this platform.

Our established deployment strategy, for which this benchmark is the foundational first step, follows a methodical, multi-stage validation process that represents a pathway to production. We will articulate this intended pathway more clearly in the paper:

**Phase 1**: Policy Development with Offline Validation and Safety Layer Development (in SustainLC). Before any live testing, operators use SustainLC to train, refine, and rigorously de-risk RL policies. The benchmark's high-fidelity nature allows for realistic pre-training and, crucially, the development of safety-critical "guardrails"—logic that prevents the agent from violating thermal or power constraints. This is a vital step that is too risky to develop on live hardware.

**Phase 2**: Hardware-in-the-Loop Validation on a Physical Testbed. The next planned phase is to validate both the digital twin's response and the trained RL controllers on a dedicated, smaller-scale physical liquid cooling testbed. This crucial hardware-in-the-loop validation provides the final layer of confidence before any interaction with production systems.

**Phase 3**: "Shadow Mode" Deployment for Trust Building. The pre-trained and hardware-vetted agent would then be deployed in a "shadow mode" in a real data center. It ingests live sensor data and computes control decisions, but these actions are only logged and compared against the existing control system, allowing operators to verify performance on real-world data without any risk.

**Phase 4**: Phased Production Integration. Following successful validation, the policy is deployed as a lightweight, inference-optimized agent (e.g., using ONNX). It integrates into the facility's control stack (e.g., BMS/SCADA), receiving live data and transmitting validated actions to hardware controllers (e.g., via BACnet/Modbus). This begins with limited, supervised control over a non-critical subset of the infrastructure, providing final validation before broader autonomous deployment.

**Table 11:** GPU utilization and power consumption for decentralized, centralized and centralized with multihead policy for control

| Metric | Decentralized Action | Centralized Action | Centralized Action with Multihead Policy |
|---|---|---|---|
| GPU Usage (Avg, peak) | (21.2 GB, 24.3 GB) | (6.4 GB, 8.2 GB) | (7.7 GB, 8.3 GB) |
| GPU Power (Avg, peak) | (436.8 W, 531.6 W) | (392.5 W, 438.3 W) | (386.4 W, 461.2 W) |

# I GPU Usage and Power Metrics

We recognize the concerns regarding the lack of quantitative analysis on memory overhead associated with batched inference. We understand that providing concrete data is crucial for evaluating the efficiency versus resource consumption. To address this, we have conducted additional experiments to measure peak memory usage in both decentralized and centralized actions, with and without batching. The results are summarized in the table 11

These results demonstrate that while decentralized actions require higher GPU usage and power, the centralized actions, particularly with the multi-head policy, show a significant reduction in both metrics. This highlights the efficiency of our approach and provides a clearer understanding of the trade-offs involved.

# J Distillation on Cabinet Policy

In case of Decision Trees, for the CT, we evaluate VIPER-based and naive uniformly weighted trees using average episode reward, $R^2$ score, and MAE against the RL (PPO) oracle in Table 12. Sample weighting yields slightly higher rewards, but predictive differences are minimal, likely because the PPO policy emphasizes wetbulb temperature for energy control. This is reflected in the decision tree's early splits, which prioritize wetbulb temperature and supply temperature, influencing the approach temperature and thus cooling power. We also show a truncated plot of the 17-depth decision tree in section J.1 generated by distilling the Cooling Tower control using [13] and explain the action preferences based on the observation variables. Overall, DT policies match RL performance with significantly lower complexity (see Table 13). Similar results for the cabinets are shown in the Appendix Tables 14 and 15 in J.

**Table 12: Distilled Tree Performance**. PPO Oracle vs. Distilled Decision Tree Policies on SLC-Gym Cooling Tower Policy w.r.t actions taken by each policy. Rewards are averaged over 10 evaluation episodes.

| Policy | Average Reward | Action Avg R2 Score | % Action Mean Absolute Error |
|---|---|---|---|
| PPO Oracle ($\pi^*$) | 397.39 | N/A | N/A |
| Distilled DT ($\pi^*$) | 370.19 | 0.937 | $0.122^oC$ |
| Naive DT (Uniform) | 365.46 | 0.8810 | $0.115^oC$ |

**Table 13: Model Complexity**. Comparison between the PPO Oracle and the Distilled Decision Tree.

| Policy | Key Complexity Metric | Value |
|---|---|---|
| PPO Oracle ($\pi^*$) | Number of Parameters Trainable | 9610 |
| Distilled DT ($\pi$) | Number of Nodes Number of Leaf Nodes Maximum Depth | 1695 848 17 |

**Table 14:** Performance Comparison of PPO Oracle and Distilled Decision Tree Policies on SLC-Gym Cabinet Blade Group Policy w.r.t actions taken by each policy. Rewards are averaged over 10 evaluation episodes.

| Policy | Average Reward | Average Action R2 Score (w.r.t Oracle Policy) | | % Action Mean Absolute Error(w.r.t Oracle Policy) | |
|---|---|---|---|---|---|
| | | CDU Controls | Blade Group Valve Controls | CDU (°C, psi) | Blade Group Valve Actions (B = 3) |
| PPO Oracle ($\pi^*$) | 1721.82 | N/A | N/A | N/A | N/A |
| Distilled DT ($\pi$) | 1695.06 | 0.902 | 0.744 | (4.36°C, 2.57psi) | (0.17, 0.28, 0.13) |
| Naive DT (Uniform) | 1537.22 | 0.783 | 0.508 | (5.64°C, 1.72psi) | (0.26, 0.35, 0.60) |

We distill the Oracle PPO reinforcement learning (RL) policy into two separate regression trees corresponding to the action groups: CDU Controls and Blade Group Valve Controls, as presented in Table 14. The decision tree for CDU Controls achieves an average reward that is marginally lower than that of the PPO Oracle. Nevertheless, it maintains a high fidelity to the original policy, as indicated by a strong $R^2$ score (0.902) and reasonably low action mean absolute errors (4.36°C, 2.57 psi).

**Table 15:** Model Complexity Comparison between the PPO Oracle and the Distilled Decision Tree.

| Policy | Key Complexity Metric | Value | | |
|---|---|---|---|---|
| | | PPO Policy | CDU Tree | Blade Group Tree |
| PPO Oracle ($\pi^*$) | Number of Parameters Trainable | 9606 | N/A | N/A |
| Distilled DT ($\pi$) | Number of Nodes | N/A | 830 | 3985 |
| | Number of Leaf Nodes | N/A | 1659 | 1993 |
| | Maximum Depth | N/A | 17 | 28 |

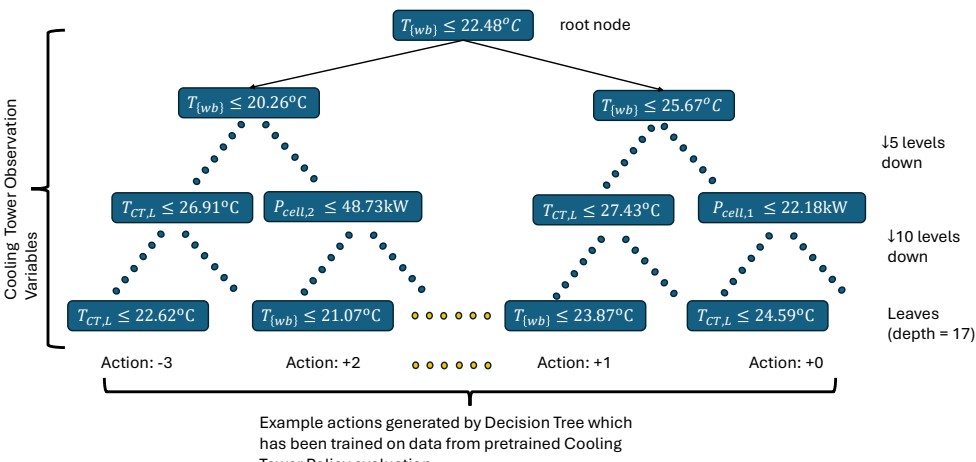

**Figure 14:** A truncated view of the Cooling Tower generated by training a Decision Tree on the pretrained Centralized Action (CA) Cooling Tower policy. The wet bulb temperature and the cooling tower water supply temperature form the most important split variables near the root as well as the leaves. The cooling tower cell power consumption variables are used to split the nodes at intermediate levels.

In contrast, the regression tree for Blade Group Valve Controls demonstrates significantly inferior performance. This degradation is largely attributed to challenges in accurately sampling from the Dirichlet distribution used to parameterize these controls. The resulting discrepancies manifest in notably lower $R^2$ scores (0.744) and increased action errors across the three blade group valves (0.17, 0.28, 0.13), as compared to the PPO Oracle. For reference, a naïve decision tree trained using uniform sampling performs even worse on both metrics, emphasizing the importance of carefully aligned sampling strategies during policy distillation.

Table 15 provides a comparative analysis of model complexity between the PPO Oracle and the distilled decision trees. The PPO Oracle, implemented as a neural policy, contains 9,606 trainable parameters. In contrast, the distilled decision trees are nonparametric models characterized by their structure: the CDU control tree comprises 830 nodes, 1659 leaf nodes, and a maximum depth of 17; the Blade Group Valve control tree is deeper and more complex, with 3985 nodes, 1993 leaf nodes, and a maximum depth of 28.

## J.1 Example of Explainable Policy

Given the substantial size and depth of the generated trees, including the complete tree visualizations is beyond the scope of this paper. However, we illustrate representative decision-making logic from the Cooling Tower control tree, which was distilled from a centralized PPO policy in Figure 14. (A complete set of tree visualizations and the corresponding generation code will be made available through a supplementary Python notebook.) The Cooling Tower decision tree exhibits a maximum depth of 17. The initial splits near the root are predominantly determined by the outside air wet bulb temperature, followed closely by the cooling tower water supply temperature (i.e., the leaving water temperature). This structure aligns with established cooling tower thermodynamic principles: the approach temperature, defined as the difference between the supply setpoint and the wet bulb temperature, is a critical factor in determining cooling effectiveness and system efficiency. These two variables dominate the decision path down to a depth of approximately 5–6 layers from the root. At

deeper levels, they remain influential, but additional features such as the power consumption of the two ($m = 2$) active cooling tower cells also begin to play a role in shaping the decision boundaries. The repeated bifurcation based on wetbulb and the water supply temperature helps decide the amount by which the supply temperature setpoint should be changed, with the power consumption at the lower levels moderating the energy consumption of the cooling tower. At the leaves, the decision making is again guided by a combination of wetbulb temperature and cooling tower water supply temperature. Overall, we realize that for fine-grained decision making on the setpoint, these two variables play a vital role, while at the intermediate level, the RL policy, as interpreted by the decision tree, tends to be guided by the average power consumption.

## K   Baseline Control Description

While ASHRAE Guideline 36 (G36) was not explicitly created for data centers, its control principles, particularly the "trim and respond" logic, can be effectively adapted to govern Coolant Distribution Unit (CDU) and Cooling Tower operations based on server temperatures, utilization metrics, and outside air parameters. This appendix details the specific implementation used as a baseline.

### K.0.1   Trim and Respond Logic for Coolant Supply Temperature

The trim and respond logic implements a reset strategy for the CDU coolant supply temperature setpoint (`COOLANT_setpoint`). It continuously "trims" the setpoint in the energy-efficient direction (warmer) when possible, but "responds" by lowering the setpoint to satisfy cooling demands when necessary.

**Initialization Parameters:**

- `COOLANT_setpoint`: Initialized to the current measured coolant supply temperature.
- `COOLANT_min`: Minimum supply temperature, set to $18\,°C$.
- `COOLANT_max`: Maximum supply temperature, set to $30\,°C$ (based on ASHRAE allowable range).
- `Trim_amount`: Amount to increase setpoint during trim phase, set to $0.1\,°C$.
- `Respond_amount`: Amount to decrease setpoint per cooling request during respond phase, set to $0.3\,°C$.
- `Request_threshold`: Minimum total requests needed to trigger the respond logic, set to 2.

**Control Sequence:**   The control sequence executes at regular intervals (typically 2-5 minutes) as shown in Algorithm 1. This approach ensures that the cooling system provides sufficient cooling to maintain required server conditions while avoiding over-cooling that wastes energy.

### K.0.2   Trim and Respond Logic for Cooling Tower Leaving Water Temperature

The cooling tower leaving water temperature setpoint (`LWT_setpoint`) control follows a similar trim and respond pattern but incorporates the outside air wet bulb temperature (`OA_wetbulb`) as a key input parameter. This optimizes free cooling potential while ensuring adequate heat rejection capacity. This cooling tower control strategy aims to maximize energy savings by raising the LWT setpoint when possible, while ensuring it remains low enough (considering the wet bulb temperature and approach) to meet the heat rejection demands indicated by the CDU system's requirements. The wet bulb reset override prevents the system from targeting an LWT that is significantly below what is efficiently achievable based on ambient conditions.

**Key Inputs and Parameters:**

- `OA_wetbulb`: Current outside air wet bulb temperature.
- `LWT_current`: Current cooling tower leaving water temperature.
- CDU Cooling Demand: Derived from the CDU control logic. G36 typically uses chiller valve positions; here, we use the proximity of `COOLANT_setpoint` to `COOLANT_min` as an indicator of high cooling demand from the CDUs.

**Algorithm 1** Coolant Supply Temperature Trim and Respond Logic

---

1: **Input:** Current server temperatures (`server_temperature`), server utilization (`server_utilization`), critical/warning temperature thresholds (`server_critical_threshold`, `server_warning_threshold`), current `COOLANT_setpoint`.
2: **Output:** Updated `COOLANT_setpoint`.

3: `total_cooling_requests` ← $0$
4: **for all** server cabinet $i$ **do**                                    ▷ Request Generation
5:     **if** `server_temperature[i]` > `server_critical_threshold` for 2 minutes **then**
6:         `requests[i]` ← $3$
7:     **else if** `server_temperature[i]` > `server_warning_threshold` for 2 minutes **then**
8:         `requests[i]` ← $2$
9:     **else if** `server_utilization[i]` > $85\%$ **and** `server_temperature[i]` is rising **then**
10:         `requests[i]` ← $1$
11:     **else**
12:         `requests[i]` ← $0$
13:     **end if**
14:     `total_cooling_requests` ← `total_cooling_requests` + `requests[i]`
15: **end for**

16:                                                        ▷ Trim and Respond Algorithm
17: **if** `total_cooling_requests` = $0$ **then**
18:     `COOLANT_setpoint` ← `COOLANT_setpoint` + `Trim_amount`        ▷ Trim towards higher temperature
19: **else if** `total_cooling_requests` ≥ `Request_threshold` **then**
20:     `COOLANT_setpoint` ← `COOLANT_setpoint` - (`Respond_amount` × `total_cooling_requests`)        ▷ Respond to requests
21: **end if**

22:                                              ▷ Ensure setpoint stays within allowable range
23: `COOLANT_setpoint` ← max(`COOLANT_min`, min(`COOLANT_max`, `COOLANT_setpoint`))

---

- `Min_approach`: Minimum achievable approach temperature (LWT - `OA_wetbulb`), set to $2.8\,°C$.

- `Optimal_approach`: Target approach temperature for efficiency, set to $3.5\,°C$.

- `LWT_max`: Maximum allowable LWT for chiller operation, set to $29.4\,°C$.

- `Trim_amount`: Amount to increase setpoint during trim phase, set to $0.1\,°C$.

- `Respond_amount`: Amount to decrease setpoint per request during respond phase, set to $0.3\,°C$.

**Initialization:**

- `LWT_setpoint`: Initialized to the current measured `LWT_current`.

- `LWT_min`: Calculated dynamically as `OA_wetbulb` + `Min_approach`.

**Control Sequence:**    The control sequence executes at regular intervals as shown in Algorithm 2.

## L   GitHub and Documentation

- Repository:

  https://github.com/HewlettPackard/sustain-lc

- Documentation Page:

  https://hewlettpackard.github.io/sustain-lc/

**Algorithm 2** Cooling Tower LWT Trim and Respond Logic

---

1: **Input:** Current `OA_wetbulb`, current `LWT_setpoint`, current `COOLANT_setpoint` (from CDU logic), `COOLANT_min`.
2: **Output:** Updated `LWT_setpoint`.

3: `total_tower_requests` $\leftarrow 0$
4:                    ▷ Request Processing from CDUs (ASHRAE G36 Addendum h adaptation)
5: **if** `COOLANT_setpoint` $< 1.05 \times$ `COOLANT_min` **then**     ▷ Triggers near 95% CDU cooling capacity
6:     `total_tower_requests` $\leftarrow 1$     ▷ Maintain request until `COOLANT_setpoint` > 1.15 * `COOLANT_min`
7: **else**
8:     `total_tower_requests` $\leftarrow 0$ ▷ Turns off below 85% CDU cooling capacity (implicit via hysteresis)
9: **end if**

10:                                  ▷ Trim and Respond Algorithm
11: **if** `total_tower_requests` $= 0$ **then**
12:     `LWT_setpoint` $\leftarrow$ `LWT_setpoint` + `Trim_amount`     ▷ Trim towards higher temperature
13: **else if** `total_tower_requests` $> 0$ **then**
14:     `LWT_setpoint` $\leftarrow$ `LWT_setpoint` - (`Respond_amount` $\times$ `total_tower_requests`)   ▷ Respond to CDU requests
15: **end if**

16:                                  ▷ Wet Bulb Reset Override
17: `LWT_optimal` $\leftarrow$ `OA_wetbulb` + `Optimal_approach`
18: **if** `LWT_setpoint` $<$ `LWT_optimal` - $0.5\,^{\circ}$C **then**
19:     `LWT_setpoint` $\leftarrow$ `LWT_optimal`     ▷ Prevent setting unrealistic targets below optimal approach
20: **end if**

21:                              ▷ Ensure setpoint stays within allowable range
22: `LWT_min` $\leftarrow$ `OA_wetbulb` + `Min_approach`     ▷ Dynamically calculated based on current conditions
23: `LWT_setpoint` $\leftarrow$ max(`LWT_min`, min(`LWT_max`, `LWT_setpoint`))

---

# M   Installation

This section provides detailed instructions for setting up the SustainLC environment.

## M.1   System Requirements

The implementation is compatible with various operating systems. All code and dependency installations were tested on macOS 15.4.1 and Ubuntu 22.04. Windows is also supported. The following prerequisites are necessary:

- Python 3.10 or higher

- Git version control system

- Command-line interface: Unix-compatible shell (bash, zsh) or PowerShell on Windows

## M.2   Installation Procedure

### M.2.1   Repository Acquisiton Procedure

The codebase must be obtained via the following commands:

```
git clone https://github.com/HewlettPackard/sustain-lc.git
cd sustain-lc
```

### M.2.2 Virtual Environment Configuration

For Unix-based or Windows systems:

```
conda env create -f environment.yml
conda activate sustain-lc
```

### M.2.3 Dependency Installation

Any further dependency installation is not required, since the environment.yml will install everything.

## M.3 Dataset Preparation

The implementation utilizes the Oakridge's cold-day heat and weather data set[3]. Researchers should download the dataset from the repository and place the processed file in the following default location:

```
data/input_04-07-24.csv
```

# N Advanced AutoCSM usage for model building

This section covers advanced topics like building custom models for LC-Opt. This requires the user to have the following repositories and software installations

## N.1 Software Installation

Dymola and OpenModelica both provide a GUI and a command-line interface (CLI) for creating, compiling, running Modelica model simulations as well as exporting them to binaries called Functional Mockup Units (FMUs).

## N.2 Repository Installation

Users need to clone the following repositories to their working folder that can be accessed by either Dymola or the OpenModelica IDEs

1. **Modelica Buildings library**: `git clone https://github.com/lbl-srg/modelica-buildings.git`
   The Modelica Buildings Library is a free, open-source library for modeling building energy and control systems, developed by Lawrence Berkeley National Laboratory. It provides comprehensive component models for HVAC systems, including heat exchangers, pumps, and valves essential for liquid cooling applications. The library enables dynamic simulation of thermal systems with fluid flow, heat transfer, and controls integration for performance analysis and optimization. Its modular architecture allows users to construct complex cooling systems by connecting components through standardized interfaces that preserve energy and mass balance. The library's extensive validation against measured data makes it suitable for accurately simulating liquid cooling systems in buildings and data centers.

2. **TRANSFORM**: `git clone https://github.com/ORNL-Modelica/TRANSFORM-Library.git`
   The TRANSFORM (TRANsient Simulation Framework Of Reconfigurable Models) Library is an open-source Modelica toolkit developed by Oak Ridge National Laboratory for modeling complex thermal-hydraulic systems. It specializes in advanced energy systems with particular strength in liquid-cooled applications, including advanced reactor designs and heat transfer loops. The library provides detailed component models for heat exchangers, pumps, compressors, and specialized fluid systems with comprehensive thermophysical property implementations. TRANSFORM excels at simulating transient behaviors in cooling systems, making it valuable for studying system responses during operational changes or upset conditions. The modular architecture enables scaling from component-level to system-level simulations with various working fluids, including specialized coolants used in high-performance liquid cooling applications.

---

[3]Available at: `https://code.ornl.gov/exadigit/datacenterCoolingModel/-/raw/main/python/data/input_04-07-24.csv?ref_type=heads`

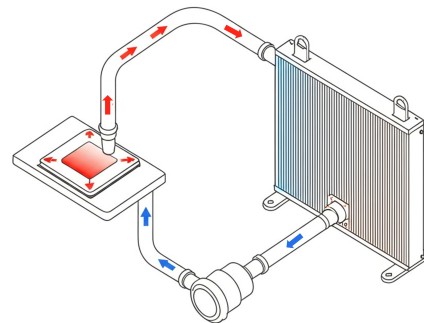

**Figure 15:** TRANSFORM for modeling thermal-hydraulic systems

3. **datacenterCoolingModel**: `git clone https://code.ornl.gov/exadigit/datacenterCoolingModel.git`

   The Data Center Cooling Model is an ORNL-developed specialized simulation framework targeting liquid cooling systems specifically for high-performance computing facilities. The repository provides detailed modeling capabilities for direct-to-chip, immersion, and rear-door heat exchanger liquid cooling technologies increasingly adopted in modern data centers. Its component models account for the complex interactions between IT equipment heat generation, coolant flow distribution, and thermal management systems at rack, row, and facility scales. The framework enables performance assessment, optimization, and efficiency analysis of cooling systems under various operating conditions and workloads. The models support integration with power consumption data to enable comprehensive energy efficiency calculations and cooling infrastructure planning for data centers.

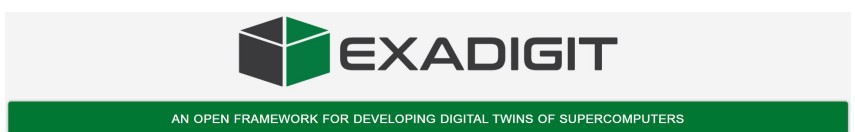

**Figure 16:** ExaDigiT supercompting consortium

4. **AutoCSM**: `git clone https://code.ornl.gov/exadigit/AutoCSM.git`

   ExaDigit AutoCSM is a template system-of-systems modeling approach for automating the development, deployment, and integration of **Cooling System Models** (CSMs) for supercomputing facilities within the **ExaDigiT framework**.

   *ExaDigiT is a digital twin of supercomputers and their thermal infrastructures. It offers insights into operational strategies, "what-if" scenarios, as well as elucidates complex, cross-disciplinary transient behaviors. It also serves as a design tool for future system prototyping. It combines telemetry and simulations, providing a virtual representation of physical systems. It supports planning, construction, and operations, offering value in decision-making, predictive maintenance, and system efficiency. In design stages, it can evaluate energy efficiency, virtually prototype cooling systems, and model network performance. During operations, ExaDigiT aids in predictive maintenance and operational optimization.*

   *ExaDigiT is built on an open software stack (Modelica, SST Macro, Unreal Engine) with an aim to foster community-driven development, we have formed a **partnership with national supercomputer centers (Oak Ridge National Laboratories, Lawrence Livermore National Labs, Los Alamos National Labs (USA), PAWSEY (Australia), LUMI (Finland), CINES (France) , CINECA (Italy), etc)** around the world to develop an open framework for modeling supercomputers.*

   AutoCSM is a Python-based framework to assist in CSM developers in accelerating the creation and deployment of system-level thermal-hydraulic CSMs. The intention is for this tool specifically to help standardize digital twin workflows for ExaDigiT. However, this tool can be used independently of ExaDigiT (and even other systems besides CSMs).

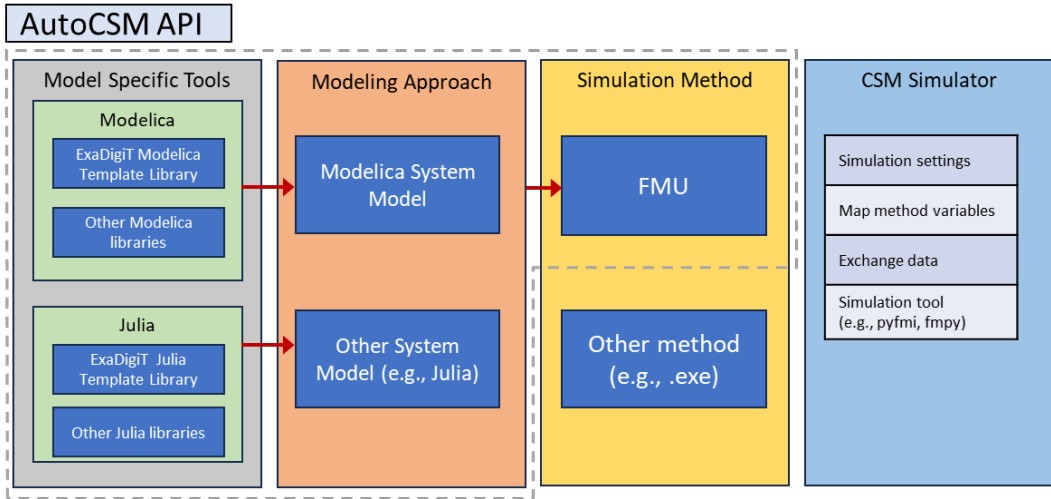

**Figure 17:** AutoCSM API in the broader ExaDigiT procedure.

Of these libraries, the user needs to access the **datacenterCoolingModel** to study the atomic structures of the thermodynamic components that can be used to build custom data center configurations. An example configuration is provided in Example JSON. This JSON describes an example hierarchical structure for the models. Further example hierarchical structures used for the results in the main paper are also included in the LC-Opt repository.

### N.3 Custom LC-Opt models using AutoCSM

The primary model building process based on the specified structure is executed by the **AutoCSM** API library. It reads the JSON file and then populates a Modelica file using elements from the datacenterCoolingModel library.

To execute this process, we simply run the

`"python run_auto_csm.py"`

from the CLI in which the JSON file and the Python files are located in the AutoCSM library. The user needs to specify the path to the desired JSON file inside the `run_auto_csm.py` file as well as compilation parameters like solver information, steps to solve etc.

The above process generates the FMU which is then wrapped inside a Gymnasium Environment for LC-Opt. Most of the common application requirements are already covered by the default Sustain-LC environment file `frontier_env.py`. If the user wishes to specify highly custom variables for logging, they have to specify those variables in the info dictionary for the environment.

## O Details of Data Center Liquid Cooling Models

### O.1 Blade Group and CDU Modeling

The thermodynamics of the blade-groups (BGs) to which the heat is added and then removed via liquid cooled plates is governed by the two equations differential equations: the heat capacitor and the conduction model. The heat capacitor $C$ represents the server's thermal storage capacity,

$$C \cdot \frac{dT}{dt} = Q_{port}(t)$$

where $T$ is the BG temperature and $Q_{port}$ is the net heat flow. $Q_{port}$ is calculated based on the heat generated by the BG based on its load heat generation rate $P_{branch}$.

$$Q_{port} = P_{branch}$$

The conduction model used to transfer heat between the server plate and the cooling liquid is represented by:

$$Q_{flow} = G_c \cdot (T_{solid} - T_{fluid})$$

where $G_c$ is the effective convection thermal conductance, dependent on coolant properties and flow rate ($m_{flow}$) and the temperature of the incoming fluid($T_{fluid}$). The overall heat transfer problem is described by

$$\Phi(Q_{port}) + Q_{flow} = C \cdot \frac{dT_{server}}{dt}$$

Server load heat generation rate $P_{branch}$ is an exogenous temporal variable determining heat generation at different temperatures ($T$). An example trace of this heat load generation is shown in input_04-07-24.csv. The primary optimization goal is to minimize energy consumption by controlling the cooling liquid flow rate $m_{flow}$ and its temperature entering the servers $T_{fluid}$. $\Phi$ is a polynomial function that helps us experimentally tune the heat generation behavior under server workloads to demonstrate the ability of RL algorithms to work with non-linear heat load generation. The quadratic term is usually adjusted to 0.015 with the linear term set at 1. The goal is to make the problem hard for RL and heuristic controllers compared to linear differential equations for heat generation.

## O.2 Cooling Tower Model

A cooling tower cools a stream of hot water by bringing it into contact with a stream of air. The primary cooling mechanisms are:

1. **Evaporative Cooling (Latent Heat Transfer)**: A small portion of the hot water evaporates into the air stream. This phase change from liquid to vapor requires a significant amount of energy (latent heat of vaporization), which is drawn from the remaining bulk water, thus cooling it. This is usually the dominant cooling effect.

2. **Sensible Heat Transfer (Convection)**: If the air is cooler than the water, there will be direct heat transfer from the water to the air due to the temperature difference. This effect is typically smaller than evaporative cooling.

The coolest temperature the water can theoretically reach is the **wet-bulb temperature** of the incoming air. The difference between the actual outlet cold water temperature and the air's wet-bulb temperature is called the **approach**.

In the Modelica models used in this work, we have three main files YorkCalc.mo for Base Empirical Correlation, coolingTower_Towb.mo for Detailed Physics and Mass/Energy Balances and CoolingTower.mo for System Integration and Fan Model.

The physics-based equations are primarily from `coolingTower_Towb.mo` and concepts in `YorkCalc.mo`). The variables used for the modeling are listed below for convenience first:

- $T_{w\_in}$: Inlet water temperature (°C or K)
- $T_{w\_out}$: Outlet water temperature (°C or K)
- $\dot{m}_{w\_in}$: Inlet water mass flow rate ( kg/s )
- $\dot{m}_{w\_out}$: Outlet water mass flow rate ( kg/s )
- $\dot{m}_{w\_evap}$: Water evaporation rate ( kg/s )
- $c_{p\_w}$: Specific heat capacity of water (J/kg.K)
- $h_{fg}$: Latent heat of vaporization of water (J/kg)
- $T_{a\_in}$: Inlet air dry-bulb temperature (°C or K)
- $T_{a\_out}$: Outlet air dry-bulb temperature (°C or K)
- $T_{wb\_in}$: Inlet air wet-bulb temperature (°C or K)
- $\phi_{a\_in}$: Inlet air relative humidity (-)
- $\phi_{a\_out}$: Outlet air relative humidity (-)
- $X_{a\_in}$: Inlet air humidity ratio ($kg_{water\_vapor}$ / $kg_{dry\_air}$)

- $X_{a\_out}$: Outlet air humidity ratio ($kg_{water\_vapor}$ / $kg_{dry\_air}$)
- $\dot{m}_a$: Dry air mass flow rate ($kg_{dry\_air}/s$)
- $h_{a\_in}$: Enthalpy of moist inlet air ($J/kg_{dry\_air}$)
- $h_{a\_out}$: Enthalpy of moist outlet air ($J/kg_{dry\_air}$)
- $U_w$: Internal energy of water in the tower (J)
- $U_a$: Internal energy of air in the tower (J)
- $m_{w\_sump}$: Mass of water in the tower sump (kg)
- $m_{a\_vol}$: Mass of dry air in the tower volume (kg)
- $m_{v\_vol}$: Mass of water vapor in the tower volume (kg)
- $Q_{tot}$: Total heat rejected by water (W)
- $Q_{sen}$: Sensible heat transfer from water to air (W)
- $Q_{lat}$: Latent heat transfer due to evaporation (W)
- $P_{fan}$: Fan power consumption (W)

### O.2.1 YorkCalc Empirical Model for Outlet Water Temperature

The core of predicting the cooling tower's performance in this model suite comes from an empirical correlation. The outlet water temperature ($T_{w\_out}$) is determined by adding an "approach temperature" ($\Delta T_{app}$) to the inlet air's wet-bulb temperature ($T_{wb\_in}$).

$$T_{w,out} = T_{wb,in} + \Delta T_{app} \tag{1}$$

The approach temperature $\Delta T_{app}$ itself is calculated using a polynomial function that depends on:

- $T_{wb\_in}$: Inlet air wet-bulb temperature.
- $R_F = \frac{\dot{m}_w}{\dot{m}_a}$: Flow ratio (water flow rate to air flow rate).
- $R_{F,nom}$: Nominal (design) flow ratio.
- $A_R$: Approach ratio at nominal conditions (a design parameter).
- $Q_{ratio} = \frac{Q_{actual}}{Q_{nominal}}$: Ratio of actual heat rejection to nominal heat rejection.

The specific polynomial form in `YorkCalc.mo` for $\Delta T_{app}$ is:

$$\begin{aligned}
\Delta T_{app} = A_R \cdot (&c_1 + c_2 T_{wb,in} + c_3 T_{wb,in}^2 + c_4 R_F + c_5 R_F^2 \\
&+ c_6 R_F T_{wb,in} + c_7 Q_{ratio} + c_8 Q_{ratio}^2 + c_9 Q_{ratio} T_{wb,in} + c_{10} Q_{ratio} R_F)
\end{aligned} \tag{2}$$

where $c_1, ..., c_{10}$ are empirical coefficients. (Note: The Modelica code simplifies this based on $use_{QRatio}$ flag. The most general form includes $Q_{ratio}$. For simplicity, we have presented a common structure; the exact terms might vary slightly based on the if conditions in the Modelica code, but the principle is a polynomial fit).

### O.2.2 Mass Balances

These equations describe how the amount of water and air (and vapor in air) changes over time within the control volume of the cooling tower. der(X) means dX/dt.

The change in water mass in the tower sump is the inlet water flow minus the outlet water flow and minus the evaporated water.

$$\frac{d(m_{a,vol})}{dt} = \dot{m}_{a,in} - \dot{m}_{a,out} \tag{3}$$

In steady-state, $\dot{m}_{w,out} = \dot{m}_{w,in} - \dot{m}_{w,evap}$. The outlet water flow is less than the inlet due to evaporation.

Assuming dry air mass flow rate is controlled and constant through the tower.

$$\frac{d(m_{a,vol})}{dt} = \dot{m}_{a,in} - \dot{m}_{a,out} \tag{4}$$

In steady-state, $\dot{m}_{a,in} = \dot{m}_{a,out} = \dot{m}_a$

The change in water vapor mass in the air inside the tower depends on vapor entering with inlet air, vapor leaving with outlet air, and water evaporating into the air.

$$\frac{d(m_{v,vol})}{dt} = \dot{m}_a X_{a,in} - \dot{m}_a X_{a,out} + \dot{m}_{w,evap} \tag{5}$$

In steady-state, $\dot{m}_{w,evap} = \dot{m}_a (X_{a,out} - X_{a,in})$. This directly links the evaporation rate to the change in air humidity.

### O.2.3 Energy Balances

The following equations describe how the energy content of the water and air changes. The change in internal energy of the water in the tower is due to enthalpy flow in, enthalpy flow out, convective heat

transfer to the air, and energy lost due to evaporation (where evaporated water carries away enthalpy $h_{w\_evap}$, often approximated as enthalpy of saturated liquid at $T_{w\_out}$).

$$\frac{d(U_w)}{dt} = \dot{m}_{w,in}h_{w,in} - \dot{m}_{w,out}h_{w,out} - Q_{sen} - \dot{m}_{w,evap}h_{w,evap} \tag{6}$$

Where $h_w = c_{p,w}T_w$ (approximately, if using a reference temperature of 0°C for enthalpy). The term $Q_{sen}$ represents sensible heat transfer. The term $\dot{m}_{w,evap}h_{w,evap}$ is closely related to $Q_{lat}$.

The change in internal energy of the air in the tower is due to enthalpy flow in, enthalpy flow out, convective heat received from water, and enthalpy gained from evaporated water vapor.

$$\frac{d(U_a)}{dt} = \dot{m}_a h_{a,in} - \dot{m}_a h_{a,out} + Q_{sen} + \dot{m}_{w,evap}h_{v,evap} \tag{7}$$

Where $h_a$ is the enthalpy of moist air (J/kg dry air), and $h_{v,evap}$ is the enthalpy of water vapor at the evaporation temperature (often taken as $T_{w\_out}$). Note that $h_a = c_{p,da}T_a + X_a(h_{fg,0} + c_{p,v}T_a)$ where $h_{fg,0}$ is latent heat at $0^oC$, and $c_{p,da}$ and $c_{p,v}$ are specific heats of dry air and vapor.

Finally, for the heat transfer calculation, we start with the total heat rejected by water (steady state). This is the primary quantity of interest from the water side.

$$Q_{tot} = \dot{m}_{w,in}c_{p,w}(T_{w,in} - T_{w,out}) \tag{8}$$

The $T_{w,out}$ here is the one determined by the YorkCalc model.

Relating Total Heat to Air Side (Steady State): This total heat rejected by the water is transferred to the air as a combination of sensible and latent heat.

$$Q_{tot} = \dot{m}_a(h_{a,out} - h_{a,in}) \tag{9}$$

This equation must hold, and it's used to find the $h_{a,out}$ (and thus $T_{a,out}$ and $X_{a,out}$)

Latent Heat Transfer:

$$Q_{lat} = \dot{m}_{w,evap}h_{fg} \tag{10}$$

where $h_{fg}$ is the latent heat of vaporization, typically evaluated at the average water temperature or $T_{w\_out}$.

Sensible Heat Transfer: The model doesn't explicitly calculate $Q_{sen}$ using a heat transfer coefficient and LMTD (Log Mean Temperature Difference) typical in detailed heat exchanger models. Instead, after $T_{w,out}$ is found from YorkCalc, and $Q_{tot}$ is known, the model solves for $\dot{m}_{w,evap}$ and the outlet air state ($T_{a,out}$, $X_{a,out}$) such that the air-side energy and mass (vapor) balances are satisfied. The $Q_{sen}$ is implicitly:

$$Q_{sen} = Q_{tot} - Q_{lat} \tag{11}$$

Or, from the air side:

$$Q_{sen} \approx \dot{m}_a c_{p,moist\_air}(T_{a,out} - T_{a,in}) \tag{12}$$

The Modelica code calculates $Q_{conv}$, which is the sensible heat transfer. It is derived from the overall energy balance once $m_{flow\_w\_evap}$ is determined.

### O.2.4 Determining Evaporation Rate and Outlet Air State

Once $T_{w,out}$ is known from YorkCalc, and thus $Q_{tot}$ is known:

- The model needs to find $\dot{m}_{w,evap}$, $T_{a,out}$, and $X_{a,out}$ (or $\phi_{a,out}$).
- This is an iterative process or a simultaneous solution. The key constraints are:
  - $\dot{m}_{w,evap} = \dot{m}_a(X_{a,out} - X_{a,in})$ (Vapor mass balance)
  - $Q_{tot} = \dot{m}_a(h_{a,out}(T_{a,out}, X_{a,out}) - h_{a,in}(T_{a,in}, X_{a,in}))$ (Air energy balance)
  - Psychrometric relations: $h_{a,out}$ depends on $T_{a,out}$ and $X_{a,out}$. Also, $X_{a,out}$ is related to $\phi_{a,out}$ and $T_{a,out}$ via saturation pressure.
  - A common assumption is that the outlet air is saturated or near-saturated at $T_{a,out}$ if the tower is efficient, but the model calculates $phi_{out\_a}$ based on $X_{out\_a}$ and $T_{out\_a}$.

The Modelica code uses `Medium.temperature_phX` and similar functions to solve for these outlet air properties based on its calculated enthalpy and humidity ratio. The `evaporationAndCondensation` record within `coolingTower_Towb.mo` attempts to model this more fundamentally, calculating $m_{flow\_Sender}$, which is $m_{flow\_w\_evap}$.

### O.2.5 Fan Power (from `CoolingTower.mo`)

The `CoolingTower.mo` model incorporates a fan model. Fan power $P_{fan}$ is typically calculated based on the air volume flow rate $\dot{V}_a$ and the pressure rise $\Delta p_{fan}$ provided by the fan, and fan efficiency $\eta_{fan}$:

$$\dot{V}_a = \frac{\dot{m}_a}{\rho_a} \quad \text{(where } \rho_a \text{ is air density)} \tag{13}$$

$$P_{fan} = \frac{\dot{V}_a \Delta p_{fan}}{\eta_{fan}} \tag{14}$$

The fan model `SpeedControlled_y` in Buildings library uses performance curves (polynomials) to relate flow rate, pressure rise, speed, and power, often normalized by nominal values. For example:

$$P_{fan} = P_{fan,nom} \cdot f(N/N_{nom}, \dot{V}_a/\dot{V}_{a,nom}) \tag{15}$$

where N is the fan speed.

# P    LC-Opt environment implementation

The environment interfaces with a high-fidelity Modelica model compiled as a Functional Mock-up Unit (FMU) version 2.0 for Co-Simulation, leveraging the `PyFMI` library for interaction.

**FMU Integration and Simulation Core**

The core of the simulation is a Modelica model representing the data center's liquid cooling thermodynamics. This model is compiled into an FMU, for example `LC_Frontier_5Cabinet_4_17_25.fmu`. The environment utilizes `PyFMI` to:

1. Load the FMU and parse its model description.
2. Instantiate the FMU for simulation.
3. Set up the experiment parameters, including start time (0.0) and a tolerance (if specified, default is FMU's choice).
4. Initialize the FMU into its starting state.
5. During an episode step:
    - Set input values (actions from the RL agent) to specified FMU variables.
    - Advance the simulation time by `sim_time_step` using the `fmu.do_step()` method. This is repeated until the agent's `step_size` is covered.
    - Get output values (observations for the RL agent and values for reward calculation) from specified FMU variables.
6. Terminate and free the FMU instance upon closing the environment or resetting for a new episode.

The FMU variable names used for interfacing are explicitly defined within the environment:

- **Action Variables:** $self.fmu\_action\_vars$ (e.g., $pump1.speed\_in$, $valve1.position\_in$)
- **Observation Variables:** $self.fmu\_observation\_vars$ (e.g., $serverRack1.T\_out$, ambient.T)
- **Power Consumption Variables (for reward):** $self.fmu\_power\_vars$ (e.g., [pump1.P, fan1.P])
- **Target Temperature Variable (for reward):** $self.fmu\_target\_temp\_var$ (e.g., $controller.T\_setpoint$)

## State (Observation) Space

The observation space is defined as a continuous `gymnasium.spaces.Box` with specific lower and upper bounds. It comprises the following variables retrieved from the FMU:

- `FrontierNode.AvgBladeGroupTemp`: Average temperature of a Blade Group in a cabinet (K).
- `FrontierNode.AvgBladeGroupPower`: Average power input to each Blade Group in a cabinet (w).

The bounds for these observations are set to e.g., `273.15 K` and e.g., `373.15 K` for temperature measurements, and between e.g., `0.0 kW` and e.g., `400 kW`, for power input measurements.

For the Cooling Tower Markov Decision Process, we have a similar observation space:

- `FrontierNode.CoolingTower.CellPower`: Average power consumption of each cell of the cooling tower (w).
- `FrontierNode.CoolingTower.WaterLeavingTemp`: Average temperature of the water leaving each cooling tower (K).
- `T_owb`: Outside air wetbulb temperature.

## Action Space

The action space is a hybrid of continuous `gymnasium.spaces.Box` and discrete `gymnasium.spaces.Discrete`, allowing the agent to control:

- `FrontierNode.CDU.Pump.normalized_speed`: Scaled speed of the CDU pump (-1 to 1).
- `FrontierNode.CDU.TempSetpoint`: Scaled Coolant supply temperature setpoint (-1 to 1).
- `FrontierNode.CDU.AvgBladeGroupValve`: Scaled Valve opening to allow coolant to collect heat from the corresponding blade group (-1 to 1).
- `FrontierNode.CoolingTower.WaterLvTSPT`: Discrete setting of cooling tower water leaving temperature setpoint delta.

These scaled values allow the neural network models used for the RL agents to learn properly and not saturate at the activation layers.

## Reward Function

The reward function guides the RL agent towards desired operational states. It is calculated at each step as:

$$R_{blade} = -\sum_{i,j} T_{i,j} \tag{16}$$

which is the negative of the aggregate temperature of the blade groups

$$R_{coolingtower} = -\sum_{i,j} P_{i,j} \tag{17}$$

which is the negative of the total cooling tower power consumption at each time step Where:

- $T_{i,j}$ is the temperature of the $j^{th}$ blade group of the $i^{th}$ cabinet $\forall\, j$ in $1 \ldots B$ and $\forall\, i$ in $1 \ldots C$
- $P_{i,j}$ is the power consumption of the $j^{th}$ cell of the $i^{th}$ cooling tower $\forall\, j$ in $1 \ldots m$ and $\forall\, i$ in $1 \ldots N$

The goal is to minimize server temperatures below the target and minimize energy consumption.

## Episode Dynamics and Simulation Control

An episode runs for a maximum of `max_episode_duration`. The agent interacts with the environment at discrete time intervals defined by `step_size`. For each agent step, the FMU's `do_step()` method is called `step_size / sim_time_step` times. The `reset()` method terminates the current FMU instance, re-instantiates and re-initializes it, ensuring a consistent starting state for each new episode. Initial observations are drawn from the FMU after initialization.

It derives some aspects of the RL problem formulation from earlier work, such as [32**?**, 33, 34].

## Q LC-Opt Training Scripts Documentation

- **Script:** `train_mh_ma_ca_ppo.py`
  This script is designed to train a Proximal Policy Optimization (PPO) agent in an environment that involves multiple agents and components. It can be configured to use Multi-Head (MH), Centralized Action (CA), and Multi-Agent (MA) features, allowing for flexible and expressive policy representations. CA implies there is a shared policy for multiple homogeneous agents within the specified environment.
  **Basic Run Command**: `python train_mh_ma_ca_ppo.py`

  **Key Configurable Parameters**

  The script uses has the following relevant parameters you can modify:

  `-exp-name (str, default: ppo_ma_ca)` Name for the experiment, used for logging.
  `-seed (int, default: 123)` Random seed for reproducibility.
  `-cuda (flag, default: True)` Enables CUDA for GPU acceleration if available. Set `-cuda False` to force CPU.
  `-env_name (str, default: MH_SmallFrontierModel)` Name of the environment
  `-agent_type (str, default: MultiHead_CA_PPO)` Type of the RL Agent
  `-max_training_timesteps (int, default: 5e6)` Total budget for training.
  `-max_ep_len (int, default: 200)` Maximum episode length.
  `-lr_actor (float, default: 3e-4)` Learning rate for the actor optimizer.
  `-lr_critic (float, default: 1e-3)` Learning rate for the critic optimizer.
  `-K_epochs (float, default: 50)` Epochs of training to run for each update.
  `-eps_clip (float, default: 0.2)` clip parameter for PPO.
  `-num_centralized_actions (int, default: 4)` Number of centralized actions for each environment.
  `-gamma (float, default: 0.80)` Discount factor for future rewards.
  `-gae_lambda (float, default: 0.95)` Lambda for General Advantage Estimation (GAE).
  `-minibatch_size (int, default: 32)` Mini-batch size for each epoch
  `-ent-coef (float, default: 0.01)` Entropy coefficient for exploration.
  `-vf-coef (float, default: 0.5)` Value function loss coefficient.
  `-num-agents (int, default: 2)` Specifies the number of agents in the custom environment.

- **Script:** `train_multiagent_ca_ppo.py`
  This script trains multiple PPO agents for a multi-agent reinforcement learning (MARL) task. Each agent has its own policy and value function, for the blade group control and the cooling tower control. It specifically employs a Centralized Action (CA) mechanism. The script is designed to work with MARL environments.
  **Basic Run Command**: `python train_multiagent_ca_ppo.py`
  The key configurable parameters for this script is identical to `train_mh_ma_ca_ppo.py`.

## R Evaluation

We provide a more user-friendly example to evaluate the agents via Jupyter notebooks. Interested users may also simply export the notebook to a Python script and run the resulting file

For evaluating the centralized action policies, users may run the `evaluate_ma_ca_ppo.ipynb` and for multihead policies, they may run `evaluate_mh_ma_ca_ppo copy.ipynb`.

## S Evaluation of Carbon Footprint for different control approaches

## T Policy Distillation using Decision Trees

To distill the policies for the pretrained agents, the users may run the `policy_distillation.ipynb` notebook

**Table 16:** Carbon footprint for different control types over a 2-day period.

| Control Type | Carbon Footprint (TonnesCO$_2$/kWh) (Cumulative over 2 days) |
|---|---|
| 1. Baseline Control | 25.24 |
| 2. CT RL + BG Baseline | 25.62 |
| 3. CT Baseline + BG RL (no valve control) | 20.95 |
| 4. CT Baseline + BG RL (with valve control) | 23.09 |
| 5. Multiagent RL | 23.37 |
| 6. Multiagent RL (Centralized Action) | 22.84 |
| 7. Multiagent RL (CA & Multihead policy) | 19.22 |

## U  Additional Results on LLMs for LC-Opt applications

We provide further examples of RL agent actions provided by different LLM agents. The LLMs are passed the input observations and the corresponding actions taken by the RL agent in evaluation, and then asked to explain with different levels of detail.

### U.1  Detailed Explanation (Llama-3.1-8B-Instruct)

---

**Detailed Explanation**

**EXECUTIVE SUMMARY:**
The current system is focused on balancing temperature stability and energy efficiency. The cabinet valves are distributed to match the power distribution across blade groups, while the cooling tower is set to very low cooling to avoid energy wastage. However, there is room for improvement in aligning temperature setpoints with goals and considering trade-offs between stability and efficiency.

**TEMPERATURE STABILITY:**
• **Overall Assessment:** The overall temperature stability is rated as moderate (score: 0.6341). While the system maintains reasonable temperature levels, there are deviations from desired goals, especially in cabinets 1 and 5.
• **Cabinet Temperature Control:** The valve positions are distributed to match the power distribution across blade groups, which helps maintain consistent temperature levels across all cabinets. However, this comes at the cost of increased energy usage compared to other configurations.
• **Cooling Tower Impact:** The cooling tower action is set to very low cooling to avoid excessive energy consumption. While this reduces the overall impact on temperature stability, it also limits the system's ability to compensate for large deviations from desired goals.

**ACTION RATIONALE:**
• **Temperature Setpoints:** The temperature setpoints do not fully align with the desired goal temperatures for each cabinet. This results in deviations from target values, especially in cabinets 1 and 5.
• **Valve Positions:** The valve positions are distributed to match the power distribution across blade groups, which helps maintain consistent temperature levels across all cabinets. However, this results in higher energy usage compared to other configurations.
• **Cooling Tower:** The cooling tower action is set to very low cooling to avoid excessive energy consumption. While this reduces the overall impact on temperature stability, it also limits the system's ability to compensate for large deviations from desired goals.

**ENERGY EFFICIENCY:**
• **Assessment:** The current actions have a moderate energy efficiency score of 0.6166. While the valve positions are distributed to match the power distribution across blade groups, this results in higher energy usage compared to other configurations. Additionally, the cooling tower action is set to very low cooling which reduces energy wastage but also limits the system's ability to compensate for large deviations from desired goals.
• **Trade Offs:** There is a trade-off between temperature stability and energy efficiency in the current actions. The valve positions are distributed to match the power distribution across blade groups, which helps maintain consistent temperature levels across all cabinets but comes at the cost of increased energy usage. The cooling tower action is set to very low cooling to avoid excessive energy consumption, which reduces the overall impact on temperature stability but also limits the system's ability to compensate for large deviations from desired goals.

**RECOMMENDATIONS:**
One potential improvement could be to adjust the temperature setpoints to better align with desired goals,

---

especially in cabinets 1 and 5. This would help reduce deviations from target values and improve overall temperature stability. Another option could be to consider alternative valve distribution configurations that balance temperature stability and energy efficiency. By optimizing both factors simultaneously, the system can achieve better results overall.
Inference Time: 4.77 seconds

## U.2 Very Detailed Explanation (Llama 3.1-8B-Instruct)

This explanation is significantly longer (approximately 1,500 words) and includes detailed scientific analysis across multiple categories. For complete response, refer to Appendix. A representative excerpt:

---

**Very Detailed Explanation**

**EXECUTIVE SUMMARY:**
The cooling system is currently maintaining a stable operating temperature across all five CDUs. This is achieved through a combination of active cooling provided by the cooling towers and passive thermal regulation via the cabinet valves. The cooling towers are operated at a very low cooling level, resulting in a slight increase in temperature compared to ideal operating conditions. The cabinet valves are adjusted to distribute cooling fluid equitably among the CDUs, balancing temperature differences while minimizing energy consumption. Overall, the system is operating within acceptable parameters and maintaining a stable thermal environment for the computational hardware.

**SYSTEM STATE ANALYSIS:**
• **Temperature Overview:**
The temperature distribution across the five CDUs is relatively uniform, with each cabinet maintaining a temperature within a narrow range. The highest temperatures are observed in CDU-5, which has experienced a higher workload and is currently operating at a temperature of 26.38 Celsius. The lowest temperatures...

• **Workload Distribution:**
The workload distribution across the five CDUs is relatively even, with each cabinet processing an approximately equal share of the total computational workload. The CDUs are configured to handle the workload in a balanced manner, ensuring that each...

• **Historical Context:**
The current system state is characterized by stable temperatures and power consumption levels, indicating that the system has been operating in a consistent and reliable manner over the past several months. The cooling system has maintained...

• **System Goals:**
The current system goals for the cooling system are to maintain a stable operating temperature across all five CDUs, while minimizing energy consumption and maximizing computational performance. The system is currently operating within...

**CABINET CONTROL ANALYSIS:**
• **Temperature Setpoints:**

  • **Description:** The temperature setpoints for the five CDUs are configured to maintain a uniform operating temperature across all cabinets. The setpoints are optimized to balance performance and energy efficiency, taking into account...

  • **Justification:** The temperature setpoints for the five CDUs are justified by the specific thermal requirements of the computational hardware. The setpoints are configured to maintain a temperature range that is optimal for...

  • **Impact Assessment:** The impact of the current temperature setpoints on system stability and performance is positive. By maintaining a uniform temperature distribution across all five CDUs, the system is able to operate...

• **Valve Distributions:**

  • **Description:** The valve distributions for the five CDUs are adjusted to balance cooling fluid flow and minimize energy consumption. The valve positions are optimized to distribute cooling fluid equitably among the cabinets...

  • **Power Alignment:** The valve distributions for the five CDUs are aligned with the power consumption distribution across the cabinets. The valve positions are adjusted to ensure that each cabinet receives an equitable...

---

- **Hydraulic Considerations:** The hydraulic considerations for the valve distributions include ensuring that the cooling fluid is distributed equitably among the cabinets, while also minimizing energy consumption...

**COOLING TOWER ANALYSIS:**
- **Action Details:** The current cooling tower action is 'very low cooling', indicating that the cooling towers are currently operating at a low cooling level to ensure that the system remains stable and efficient. This action is justified...
- **Thermal Dynamics:** The thermal dynamics of the cooling towers are characterized by a low cooling level, resulting in a slight increase in temperature compared to ideal operating conditions. This is justified by the specific thermal...
- **Weather Interactions:** The current weather conditions are not directly influencing the cooling tower action. The cooling towers are adjusted to maintain a uniform temperature distribution across all five CDUs, ensuring that...

**ENERGY EFFICIENCY ANALYSIS:**
- **Component Efficiency:**

  - **Cooling Tower:** The cooling tower efficiency is relatively low, resulting in a slight increase in temperature compared to ideal operating conditions. This is justified by the specific thermal requirements...

  - **Pumps And Valves:** The pump and valve efficiency is relatively high, resulting in minimal energy consumption. The valve positions are adjusted to ensure that each cabinet receives an equitable share...

  - **Secondary Cooling:** The secondary cooling loop efficiency is relatively high, resulting in minimal energy consumption. The valve positions are adjusted to ensure that each cabinet receives an equitable...

- **Optimization Strategy:** The current optimization strategy involves balancing energy efficiency and system stability. The temperature setpoints are configured to maintain a uniform operating temperature across all cabinets...
- **Performance Metrics:** The performance metrics for the cooling system include temperature stability, energy efficiency, and workload distribution. The temperature stability is relatively high, with minimal fluctuations...

**THERMODYNAMIC IMPLICATIONS:**
- **Heat Transfer:** The heat transfer dynamics within the cooling system are characterized by the distribution of cooling fluid through the cabinet valves, ensuring that each cabinet receives an equitable share of the cooling fluid...
- **Thermal Gradients:** The thermal gradients across the five CDUs are minimal, with each cabinet maintaining a uniform temperature distribution. The valve positions are adjusted to minimize thermal gradients, ensuring that...
- **Fluid Dynamics:** The fluid dynamics within the cooling system are characterized by the distribution of cooling fluid through the cabinet valves, ensuring that each cabinet receives an equitable share of the cooling fluid...

**RECOMMENDATIONS:**
- **Short Term:** The short-term recommendations for the cooling system include monitoring temperature and workload distributions, adjusting valve positions as needed, and optimizing the Cooing tower action...
- **Long Term:** The long-term recommendations for the cooling system include ongoing monitoring of temperature and workload distributions, regular adjustments to valve positions as needed, and continued optimization...
- **Trade Off Analysis:** The trade-off analysis for the cooling system involves balancing system stability and energy efficiency. The current configuration and operating conditions of the cooling system are optimized to maintain...

**SCIENTIFIC INSIGHTS:**
The scientific insights from the current system behavior include the optimal temperature setpoints for the five CDUs, the appropriate valve positions to ensure equitable cooling fluid distribution, and the ideal cooling tower action to maintain stability and efficiency. These insights are derived from a thorough analysis of the system's performance metrics, including temperature stability, energy efficiency, and workload distribution.
Inference Time: 15.25 seconds

## U.3 Qwen 3-8B Short Explanation

> **Short Explanation**
>
> **EXPLANATION:**
> The cooling system adjusted valve positions across cabinets to balance cooling and energy efficiency. Cooling tower operation was set to very low cooling due to stable temperature trends.
> **TEMPERATURE IMPACT:**
> • **Overall:** Temperatures remain within the normal operating range with minor deviations.
> • **Cooling Tower:** Very low cooling reduces system cooling capacity, which may limit temperature control effectiveness.
> **ENERGY EFFICIENCY IMPACT:**
> Very low cooling tower operation contributes to moderate energy efficiency.
> Inference Time: 1.11 seconds

## U.4 Qwen 3-8B Detailed Explanation

> **Detailed Explanation**
>
> **EXECUTIVE SUMMARY:**
> The cooling system is maintaining stable temperatures with a focus on energy efficiency, but some cabinets are not meeting their temperature goals. Cooling tower action is at a very low level, and valve positions are evenly distributed across blade groups to balance power consumption.
> **TEMPERATURE STABILITY:**
> • **Overall Assessment:** The system is maintaining stable temperatures with a stability score of 0.634, indicating moderate control effectiveness. While most cabinets are within the normal operating range, some deviations suggest room for improvement.
> • **Cabinet Temperature Control:** Valve positions are evenly distributed across blade groups (each group receiving ∼33.3% of cooling), which helps maintain consistent temperature distribution. However, the current cooling tower action is at a very low level, limiting overall cooling capacity and causing some cabinets to exceed their temperature goals.
> • **Cooling Tower Impact:** The very low cooling tower action (category 0) is significantly restricting cooling capacity, leading to higher than desired temperatures in some cabinets. This suggests that increasing cooling tower activity could improve temperature stability, though it would likely increase energy consumption.
> **ACTION RATIONALE:**
> • **Temperature Setpoints:** Temperature setpoints are aligned with the normal operating range (293.15K-313.15K). Cabinet goals vary, with some cabinets having lower goals (e.g., cdu-cabinet-1 at 20.7°C) and others having higher goals (e.g., cdu-cabinet-5 at 39.9°C). The current temperatures are mostly within this range, but some cabinets, like cdu-cabinet-5, are significantly below their goals, indicating potential undercooling or incorrect setpoint alignment.
> • **Valve Positions:** Valve positions are evenly distributed across blade groups (each group receiving ∼33.3% of cooling), which ensures balanced cooling across all components. This distribution matches the power distribution across blade groups, maintaining thermal equilibrium and preventing localized overheating.
> • **Cooling Tower:** The cooling tower is set to a very low cooling action (category 0), which minimizes energy consumption but also limits cooling capacity. This decision prioritizes energy efficiency but results in some cabinets not reaching their temperature goals, particularly those with higher cooling requirements.
> **ENERGY EFFICIENCY:**
> • **Assessment:** Energy efficiency is moderate, with a score of 0.616. The low cooling tower action directly contributes to energy savings, but this comes at the cost of reduced cooling capacity and potential temperature instability.
> • **Trade Offs:** The current actions prioritize energy efficiency by limiting cooling tower activity, which results in some cabinets being cooler than their goals while others are warmer than desired. This trade-off between energy savings and temperature stability needs to be balanced for optimal system performance.
> **RECOMMENDATIONS:**
> Consider increasing cooling tower activity to improve temperature stability, especially for cabinets with higher cooling requirements. Additionally, review and adjust temperature setpoints to ensure they align with actual operational needs and system capabilities.
> Inference Time: 4.91 seconds

## U.4.1 Qwen 3-8B Very Detailed Explanation

**Very Detailed Explanation**

**EXECUTIVE SUMMARY:**
The cooling system shows stable temperature deviations with minimal fluctuations, indicating consistent cooling performance. The cooling tower is operating at very low cooling capacity (action 0), which is likely causing reduced cooling efficiency. Cabinet temperatures are below their respective goals, particularly cabinet 5, which has a significant deviation of -13.62K, suggesting potential overcooling.

**SYSTEM STATE ANALYSIS:**

● **Temperature Overview:**
Cabinet temperatures are predominantly below their setpoints, with cabinet 5 showing the most significant deviation (-13.62K) at 26.38°C versus a goal of 39.99°C. Cabinet 2 and 4 also exhibit notable deviations (-7.93K and -7.81K, respectively), while cabinets 1, 3, and 5 are significantly below targets. This suggests that the cooling system is overcooled...

● **Workload Distribution:**
Power distribution across cabinets is uniform, with each cabinet receiving approximately 33.33% of the total power. This equal distribution indicates a balanced computational workload, which is essential for maintaining thermal stability...

● **Historical Context:**
The system has maintained stable temperature deviations and efficiency scores over time, indicating consistent performance. However, the current low cooling tower operation (action 0) suggests a potential shift towards energy conservation...

● **System Goals:**
Current conditions deviate from the system's primary goal of maintaining optimal temperature while minimizing energy consumption. The overcooling of cabinets, particularly cabinet 5, indicates that the cooling tower's low operation is not effectively...

**CABINET CONTROL ANALYSIS:**

● **Temperature Setpoints:**

- **Description:** The temperature setpoints for each cabinet are set to specific target temperatures, with cabinet 5 having the highest goal of 39.99°C. The current cabinet temperatures are significantly below these setpoints...

- **Justification:** The setpoints are determined by the required thermal management for each cabinet's workload. Higher power cabinets (e.g., cabinet 5) have higher temperature goals to accommodate increased heat generation...

- **Impact Assessment:** Overcooling leads to increased energy use and operational costs, while also potentially causing unnecessary wear on cooling components. The system's stability is maintained, but the energy efficiency score...

● **Valve Distributions:**

- **Description:** Valve positions are uniformly distributed across cabinets, with each cabinet having a similar proportion of cooling resources allocated. This uniform distribution suggests a strategy of equal cooling...

- **Power Alignment:** The uniform valve distribution aligns with the balanced power distribution across cabinets, ensuring that each cabinet receives equivalent cooling capacity. This approach is suitable for systems...

- **Hydraulic Considerations:** The uniform valve distribution ensures consistent coolant flow across all cabinets, which is important for maintaining hydraulic balance in the cooling system. However, this approach may lead...

**COOLING TOWER ANALYSIS:**

● **Action Details:** The cooling tower is operating at action level 0, which corresponds to very low cooling capacity. This action decreases the cooling tower's heat rejection capacity, potentially leading to reduced cooling efficiency...

● **Thermal Dynamics:** The low cooling tower operation results in reduced heat rejection, which means the system is relying more on internal cooling mechanisms. This could lead to increased temperatures in cabinets...

● **Weather Interactions:** The cooling tower's performance is influenced by ambient temperature and humidity levels. If the ambient temperature is high, the cooling tower's effectiveness decreases, but the current low operation...

**ENERGY EFFICIENCY ANALYSIS:**

● **Component Efficiency:**

- **Cooling Tower:** The cooling tower's efficiency is reduced due to its operation at very low cooling capacity (action 0). This leads to higher energy consumption for the same cooling effect, decreasing overall system efficiency...

- **Pumps And Valves:** Pump energy consumption is likely stable due to the uniform valve distribution and consistent flow rates. Valves are operating within their efficient range, but the uniform distribution may not be optimal...

- **Secondary Cooling:** The secondary cooling loop is operating efficiently, as evidenced by the stable temperature deviations and consistent power distribution. The secondary cooling system is effectively managing...

• **Optimization Strategy:** The current strategy prioritizes energy conservation by operating the cooling tower at very low capacity. This reduces energy costs but may compromise temperature stability. An optimized strategy would involve...

• **Performance Metrics:** The system's energy efficiency score is 0.6166, indicating room for improvement. The cooling tower's low operation contributes to this score, as it is not effectively managing the thermal load...

**RECOMMENDATIONS:**

• **Short Term:** Increase cooling tower operation to a moderate level to improve heat rejection capacity and reduce unnecessary overcooling. Adjust valve positions to provide more targeted cooling to cabinets with higher thermal loads...

• **Long Term:** Implement a dynamic cooling tower control strategy based on real-time thermal loads and ambient conditions. Introduce variable valve distribution to match cooling resources with individual cabinet demands...

• **Trade Off Analysis:** Increasing cooling tower operation improves temperature stability but may increase energy costs. Adjusting valve positions for targeted cooling improves efficiency but requires more complex control strategies...

**SCIENTIFIC INSIGHTS:**

The system's performance highlights the importance of dynamic thermal management in data center cooling. Overcooling, while maintaining stability, leads to increased energy consumption and potential inefficiencies. The uniform valve distribution and low cooling tower operation suggest a conservative approach to energy management, which may not be optimal for long-term sustainability.

Inference Time: 13.49 seconds

