# OpenReview forum: "LC-Opt: Benchmarking Reinforcement Learning and Agentic AI for End-to-End Liquid Cooling Optimization in Data Centers"
_NeurIPS.cc/2025/Datasets_and_Benchmarks_Track — NeurIPS 2025 Datasets and Benchmarks Track poster_

### Official Review · Reviewer_5Eiq · 2025-06-29

**Rating:** 4
**Confidence:** 2

**Summary:**

This paper introduces SustainLC, a reinforcement learning benchmark platform designed for optimizing energy efficiency in liquid-cooled high-performance data centers. The platform supports hierarchical control tasks across diverse control structures. The authors implement fine-grained control interfaces and state-space decomposition mechanisms, and employ a multi-head policy architecture to enhance responsiveness and efficiency in coolant allocation. Experimental results show that SustainLC consistently outperforms industry-standard baselines in both temperature regulation and energy efficiency. Furthermore, the integration of policy distillation and large language model–based explanation modules significantly improves the interpretability and practical applicability of the control strategies.

**Dataset Code Accessibility:**

Yes

**Ethical Considerations:**

No, there are no or only very minor ethics concerns

**Final Justification:**

I keep my original rating.

**Limitations Weaknesses:**

1. The paper mentions that batched inference improves inference efficiency, but does not provide concrete data on memory overhead, lacking quantitative analysis of its efficiency versus resource consumption.
2. The experiments focus on a specific data center configuration, lacking comprehensive evaluation under diverse weather conditions and varying AI workload patterns. Its generalizability and practical feasibility in other real-world in other real-world data centers settings remain to be demonstrated.

**Strengths Contributions:**

1. SustainLC is presented as an end-to-end, customizable, and scalable benchmark platform supporting multi-level control tasks from blade groups to cooling towers. Its modular structure and FMU-based export mechanism ensure strong generality and extensibility.
2. The paper provides a systematic description of the modeling, control problem formulation, reinforcement learning design, and explainability modules.

---

> ### Author Rebuttal · Authors · 2025-07-30
>
> ### Rebuttal to Reviewer 5Eiq
>
> We would like to express our gratitude to Reviewer 5Eiq for the thorough review and valuable feedback on our submission. We appreciate the  acknowledgment of the key strengths of SustainLC of end-to-end design, scalability, systematic methodology, and novel explainability features.and would like to address the limitations highlighted.
>
> ### GPU Usage and Power Metrics
>
> We recognize the concerns regarding the lack of quantitative analysis on memory overhead associated with batched inference. We understand that providing concrete data is crucial for evaluating the efficiency versus resource consumption. To address this, we have conducted additional experiments to measure peak memory usage in both decentralized and centralized actions, with and without batching. The results are summarized in the table below:
>
> | Metric ↓                 | Decentralized Action     | Centralized Action     | Centralized Action with Multihead Policy |
> |--------------------------|---------------------------|--------------------------|-------------------------------------------|
> | GPU Usage (Avg, peak)    | (21.2 GB, 24.3 GB)         | (6.4 GB, 8.2 GB)          | (7.7 GB, 8.3 GB)                           |
> | GPU Power (Avg, peak)    | (436.8 W, 531.6 W)         | (392.5 W, 438.3 W)        | (386.4 W, 461.2 W)                         |
>
> These results demonstrate that while decentralized actions require higher GPU usage and power, the centralized actions, particularly with the multi-head policy, show a significant reduction in both metrics. This highlights the efficiency of our approach and provides a clearer understanding of the trade-offs involved.
>
> ### Addressing Generalizability
>
> The primary goal of this foundational paper is to introduce a sustainability benchmark with an example of a high fidelity design of a real data center at scale. One of the best high fidelity end-to-end liquid cooling model to our knowledge is the Frontier supercomputer digital twin of Oak Ridge from the ExaDigiT consortium of supercomputer users.
> However, our design focuses on enabling generalizability in two crucial fronts.
>     1. Firstly, it enables easy customization of weather and workload patterns across global geographic locations.
>     2. The open-source, modular design allows users to customize the reference data center model and implement their own data center configurations for benchmarking, design guidance, diagnostics, and digital twin applications.

---

> > ### Author Response · Authors · 2025-08-04
> >
> > Thank you again for your thoughtful review and for acknowledging our rebuttal. We noticed that your review **currently does not display a rating**. Since your **earlier feedback included a positive rating** with comments, we would appreciate it if you could kindly confirm or resubmit your rating to ensure the review record is complete.

---

> > ### Comment · Reviewer_5Eiq · 2025-08-05
> >
> > My concerns have been addressed well.

---

### Official Review · Reviewer_1xyc · 2025-07-01

**Rating:** 5
**Confidence:** 2

**Summary:**

SustainLC introduces a reinforcement learning (RL) benchmark for end-to-end control of liquid cooling systems in sustainable high-performance computing (HPC) data centers. Based on a high-fidelity digital twin of the Frontier supercomputer, it provides detailed thermofluidic modeling and supports multi-agent RL, centralized/decentralized control, and explainable AI via policy distillation and LLM-based natural language explanations. The environment is open-sourced with full documentation.

**Dataset Code Accessibility:**

Yes

**Dataset Code Comments:**

Well-documented GitHub repository with code, FMUs, and training/evaluation scripts.

Permissive licensing (Apache 2.0).

**Ethical Considerations:**

No, there are no or only very minor ethics concerns

**Final Justification:**

My Concern has been resolved, and I will increase the rating.

**Limitations Weaknesses:**

- Limited Scale Testing: Experiments are conducted on small-scale setups; scalability and computational overhead need further analysis.

- Distillation Challenges: Policy distillation performance degrades for Dirichlet-distributed actions.

- Synthetic Workloads: Real-world telemetry and noise injection would strengthen generalization claims.

- LLM Evaluation Gaps: LLM-based explanations lack quantitative validation (e.g., coherence, faithfulness).

**Strengths Contributions:**

- Novelty: First comprehensive benchmark combining high-fidelity modeling with scalable RL control for liquid-cooled HPC systems.

- Practical Relevance: Enables fine-grained thermal regulation and energy optimization at multiple levels (cooling tower, cabinet, blade group).

- Explainability: Integrates decision trees and LLMs to enhance operator trust and policy transparency.

- Open Access: Fully open-source with reproducible experiments, FMU models, and Gymnasium interface support.

---

> ### Author Rebuttal · Authors · 2025-07-30
>
> ### Response to Reviewer 1xyc
> We acknowledge the limitations you highlighted and appreciate the opportunity to clarify our contributions and address your concerns.
>
> 1. **Scalability Concerns**: We understand your concerns regarding the scale of our experiments. To clarify, our scalability experiments presented in Table 4 evaluate configurations up to 25 cabinets, which represents a significant portion of a data hall. While our initial ablations in Table 3 utilize a 5-cabinet setup, we explicitly test the scalability of our best-performing agent in Table 4: Performance on Scale. The results demonstrate that our RL policy not only maintains its performance but often widens its advantage over the ASHRAE G36 baseline as the system size increases. For instance, in the 25-cabinet configuration, our policy reduces average cooling tower power consumption by nearly 20% (from 1381.92 kW to 1109.84 kW) while achieving significantly better temperature compliance (92.61% vs. 83.08%). We will revise the text to make the scale of our experiments clearer and add a sentence to the main text to highlight these specific results for the reader.
>
> 2. **Distillation Challenges**: We agree that distilling a policy that outputs a complex, constrained distribution (like Dirichlet) into a regression tree is a non-trivial problem. We believe this finding is significant and have reported it transparently in Appendix A.7 (lines 399-406) and Table 13. Apologize that we did not mention this in the main paper and will move it in to the main paper for camera ready if it is accepted
>
>     By highlighting this challenge, our benchmark serves as a concrete test case for the community to develop more advanced policy distillation techniques that can better handle complex, constrained action spaces. We view this not as a flaw but as a demonstration of the benchmark's value in surfacing critical research problems.
>
> 3. **Synthetic Workloads**: In Table 3, we mention the use of an "unseen exogenous trace," which incorporates real-world workload data and weather information from ORNL's power consumption and weather data. We are committed to enhancing our evaluation by running tests with "noise injection" to the data and will report the results in our revisions.
>
> 4. **LLM Evaluation Gaps**: We are actively exploring methods to evaluate and ensure the alignment of LLM rationale with system dynamics. We believe that releasing this benchmark code and engaging with the ExaDigiT community of supercomputer users will be crucial in scaling this effort. The inclusion of the LLM component is intended as a proof-of-concept to demonstrate a promising future direction, rather than a production-ready solution.
>
> We hope that these clarifications reflect the technical solidity of our work and may help reassess its impact on enabling the ML community to join forces on data center sustainability.

---

> > ### Comment · Reviewer_MBb9 · 2025-08-07
> >
> > The author's rebuttal has addressed most of my concerns, except for the real-world validation and explainability evaluation, corresponding to authors’ first and fourth rebuttal comments.
> >
> > R1 - The benchmark uses a high-fidelity digital twin, but the RL policies have not been tested in real systems. There is still a clear gap between simulation and real-world use. The paper would be stronger with a clear discussion on how to close this gap or how the trained RL methods could be used in a wider range of real-world data centers. Such a discussion can also emphasize the ability of the proposed benchmark to train more generic liquid cooling control RL algorithms.
> >
> > R4 - The LLM module is still at an early stage. While the added examples help, there is no formal evaluation of how accurate or useful the explanations are. The planned benchmark is a good direction, but even basic user studies or analysis would make the current claims stronger.
> >
> > Considering these issues, I will keep my current score.

---

> > ### Author Response · Authors · 2025-08-08
> >
> > We thank the reviewer again for the continued engagement and for acknowledging our other revisions. We would like to provide clarifications regarding points R1 and R4 to correct potential misunderstandings that may otherwise influence the final decision.
> >
> > *"Such a discussion can also emphasize the ability of the proposed benchmark to train more generic liquid cooling control RL algorithms."*
> >
> > ### Generalizability of Sustain-LC for Diverse Cooling Architectures
> >
> >
> > ### Generalization capabilities in Sustain-LC via **AutoCSM API to build different liquid cooling models**
> >
> > A core design principle of SustainLC is its inherent generalizability. The Frontier model represents a comprehensive, liquid cooling design, from which a wide variety of conventional data center cooling architectures can be derived.
> >
> > Our proposed liquid cooling benchmark is highly customizable via the **AutoCSM API**, which we have highlighted in **Figure 1 (part 1)** and in **Section 3.2 (item 1)** in the main paper. We also provided (at the time of initial submission) in our  **GitHub landing page** a very detailed description on the top of generalization and customizability under the **"Custom Sustain-LC models using AutoCSM"** heading for the **"Advanced AutoCSM usage for model building"**. The user will be able to design different data centers with the provided instructions simply by editing the configurations in the JSON file (provided under the *exadigit/datacenterCoolingModel/data/input_specification_frontier_test.json* file). Currently, the customization will allow the user to change
> >
> > 1. Different IT Rack design configurations
> > 2. Different individual Server Settings configurations in each IT Cabinet
> > 3. The model specifications of the Cooling Tower fans, including their power flow characteristics
> > 4. The model specifications of the Heat Exchangers and Heat Reuse Components
> > 5. Number of IT Servers and Cooling Distribution Units and the coolant flow arrangements in either "series" or "parallel" configurations
> > 6. Number of pumps for the cooling, high temperature, cooling tower loops
> > 7. The arrangement and number of water or coolant flow between the cooling tower and IT CDU-Cabinet loops
> >
> > ### Generalization for RL agents
> >
> > Given any user configuration from above, Sustain-LC will automatically set up the environment that is completely OpenAI Gym compatible. Hence, the user can train the RL agents from our provided library of agents or any other popular libraries like **sb3, rllib**. The end user just needs to specify the control action spaces and the RL algorithms that will be compatible with the designed data center.

---

> > ### Author Response · Authors · 2025-08-08
> >
> > *"how the trained RL methods could be used in a wider range of real-world data centers. "*
> >
> > ### Bridging Simulation-to-Real
> >
> > We thank the reviewer for underscoring the importance of the simulation-to-reality pathway, a topic that is central to our work’s motivation. The digital twin at the core of SustainLC is not a theoretical model; it is a high-fidelity simulation that has been validated against the operational dynamics of the Frontier supercomputer's cooling system. This validation provides the confidence needed for its use in developing next-generation control strategies.
> >
> > The primary scope of a benchmark is to provide a standardized, accessible, and risk-free environment for developing and comparing novel control strategies—something that is infeasible and cost-prohibitive on a live, multi-million dollar production supercomputer. SustainLC provides exactly this platform.
> >
> > **Our established deployment strategy, for which this benchmark is the foundational first step, follows a methodical, multi-stage validation process** that represents a pathway to production. We will articulate this intended pathway more clearly in the paper:
> >
> > **Phase 1: Policy Development with Offline Validation and Safety Layer Development** (in SustainLC). Before any live testing, operators use SustainLC to train, refine, and rigorously de-risk RL policies. The benchmark's high-fidelity nature allows for realistic pre-training and, crucially, the development of safety-critical "guardrails"—logic that prevents the agent from violating thermal or power constraints. This is a vital step that is too risky to develop on live hardware.
> >
> > **Phase 2: Hardware-in-the-Loop Validation on a Physical Testbed**. The next planned phase is to validate both the digital twin's response and the trained RL controllers on a dedicated, smaller-scale physical liquid cooling testbed. This crucial hardware-in-the-loop validation provides the final layer of confidence before any interaction with production systems.
> >
> > **Phase 3: "Shadow Mode" Deployment for Trust Building**. The pre-trained and hardware-vetted agent would then be deployed in a "shadow mode" in a real data center. It ingests live sensor data and computes control decisions, but these actions are only logged and compared against the existing control system, allowing operators to verify performance on real-world data without any risk.
> >
> > **Phase 4: Phased Production Integration**. Following successful validation, the policy is deployed as a lightweight, inference-optimized agent (e.g., using ONNX). It integrates into the facility's control stack (e.g., BMS/SCADA), receiving live data and transmitting validated actions to hardware controllers (e.g., via BACnet/Modbus). This begins with limited, supervised control over a non-critical subset of the infrastructure, providing final validation before broader autonomous deployment.
> >
> > This structured approach is a necessary roadmap for the community. SustainLC is one of the first comprehensive **open source** benchmarks designed to facilitate this global effort. It empowers researchers to make liquid cooling more energy-efficient and enables the high-density GPU integration required for the AI infrastructure of the future.
> >
> > To make this explicit, we would add this description to the final paper or the appendix. **We thank the reviewer for bringing this to our attention.**

---

> > ### Author Response · Authors · 2025-08-08
> >
> > *"The LLM module is still at an early stage. While the added examples help, there is no formal evaluation of how accurate or useful the explanations are. The planned benchmark is a good direction, but even basic user studies or analysis would make the current claims stronger."*
> >
> > ### LLM guided Action Explanation, Guidance and Validation and Examples
> >
> > To address the need for more formal evaluation of the LLM-based explanations and guidance, this section provides concrete examples from our experiments. Each example includes the system observation, the action proposed by the RL agent, a natural language explanation generated by the LLM, and a validity assessment from a human data-center expert.
> >
> > These examples are designed to showcase the LLM’s ability to:
> >
> > * Correctly interpret the RL agent’s actions based on system state.
> > * Provide plausible reasoning for the control decisions.
> > * Identify suboptimal or inefficient actions from the RL agent.
> > * Offer actionable guidance to improve safety and energy efficiency.
> >
> > They are **provided under the sustain-lc GitHub Readme as well as the sustain-lc documentation page** at "Agentic LLM-Based Digital Twin for Liquid Cooling/LLM Explanation and Guidance Evaluation" **subsection** under the **Sustain-lc documentation**. We provide a few of these examples here for the reviewer's convenience
> >
> > ### Example 1: LLM Guided Explanation and Guidance with Validation
> >
> > *Obs*: (307.2, 306.74, 309.77, 21567.55,  44779.22, 33672.41)
> > *Act (from RL agent)*: (0.19, 0.61, 0.2, 22.39, 25.52)
> >
> > *Explanation*: Branch 2 remains relatively cool, allowing the controller to redistribute cooling capacity toward more loaded or hotter branches. A setpoint of 22.39 and a controlled flow of 25.52 signify this adaptive behavior. The decision not only maximizes performance-per-watt but also aligns with global decarbonization efforts. It is recommended to reduce branch 1 coolant allocation to ~0.1 and to allocate the rest to branch 3 since it has a higher workload.
> >
> > *Validity (Human Expert Evaluation)*: “This is one of the better examples where the agent is explaining the actions for all three blade groups. It however, provides a good way of refining the RL valve actions.”
> >
> > ### Example 2: LLM Guided Explanation Only with Validation
> > *Obs*: (315.45, 314.04, 311.25, 46806.57, 46806.57, 31632.4)
> >
> > *Act (from RL agent)*: (0.24, 0.35, 0.41, 40.95, 24.66)
> >
> > *Explanation*: Given that thermal readings remain within tolerable bounds, the agent increases the setpoint to 40.95 for energy conservation. Coolant flow is distributed with branch 3 receiving the most, targeting demand while sidestepping unnecessary cooling. This approach supports environmental compliance standards while ensuring uptime. Reducing cooling overheads has cascading social benefits, especially in energy-constrained regions.
> >
> > *Validity (Human Expert Evaluation)*: “While the LLM response correctly attributes the increased temperature setpoint of the coolant due to moderate temperatures in the cabinet, it does not completely explain the other values that were generated by the reinforcement learning agent. Also, it does not explain why the current distribution of the fluid happens across the three branches.”

---

> > ### Author Response · Authors · 2025-08-08
> >
> > "The LLM module is still at an early stage."
> >
> > ### LLM-RL guided improved control
> > Based on the experiments above, we also plan to incorporate into the Sustain-LC GitHub our latest progress on using LLM-RL-enabled hybrid control. The following table shows the comparison between different standalone RL, LLM or hybrid control paradigm, where we track the average blade group (BG) temperature, their values within certain bounds (%Steps within 20-40C), and the Cooling Tower Power consumption
> >
> >
> > This table presents a performance comparison of various control strategies, including Large Language Model (LLM) and LLM-Reinforcement Learning (RL) hybrids. The evaluation is based on average blade temperature, cooling tower power consumption, and the percentage of time the temperature remained within the 20-40°C range. According to the experiment's criteria, optimal performance is indicated by lower blade temperatures and cooling tower power, alongside a higher percentage of steps within the specified temperature range. The experiment was conducted using a single tower with two cells, five cabinets each containing three blade groups, and was tested against workload data profile 2 from ORNL in the ExaDigiT library.
> >
> > | Method | Avg. Blade Temp (C) | Cooling Tower Power (W) | % Steps 20–40° |
> > | :--- | :--- | :--- | :--- |
> > | ASHRAE (Baseline) | 32.26 | 26731.31 | 76.92 |
> > | RL | 30.65 | 24131.52 | 93.28 |
> > | LLM Base (LLaMA) + Few-shot | 31.42 | 26500.00 | 91.25 |
> > | LLM Base (Qwen) + Few-shot | 30.99 | 25100.00 | 93.75 |
> > | LLM (LLaMA Fine Tuned by RL traces) | 30.20 | 24001.14 | 95.33 |
> > | LLM (Qwen Fine Tuned by RL traces) | 30.01 | 23378.22 | 95.97 |
> > | LLM (LLaMA FT + Few-shot) | 29.72 | 24689.92 | 95.12 |
> > | LLM (Qwen FT + Few-shot) | 29.37 | 23750.74 | 96.08 |
> > | LLM (Qwen FT) → RL Hybrid | 30.31 | 26759.77 | 93.97 |
> > | RL → LLM (Qwen FT) Hybrid | 28.29 | 23371.92 | 96.80 |
> >
> >
> >
> > ### Future Work LLM work on Sustain-LC
> >
> > We also aim to incorporate further metrics to enhance our understanding of how controller actions contribute to the following key performance indicators using LLMs:
> >
> > - **Power Efficiency**: Calculated as IT output divided by the sum of Cooling and IT input.
> > - **Temperature Stability**: Monitoring fluctuations to ensure optimal operating conditions.
> > - **Carbon Footprint**: Utilizing CO2 intensity data to assess the sustainability of decisions made by the RL agent.
> > - **Confidence Score**: Providing a quantitative measure for human evaluation of the RL agent's decisions.
> >
> > SustainLC provides a pioneering benchmark/framework that synergizes Reinforcement Learning (RL) with Large Language Model (LLM) to deliver sustainable and interpretable control for liquid-cooled data centers.

---

> > ### Author Response · Authors · 2025-08-08
> >
> > We believe these additions may address your concern and **clarify the primary role and contribution** of our work for generalizability, sim-to-real bridge, and concrete examples on validity and evaluations of LLMs for explainability and performance in the context of the NeurIPS benchmark track. Thank you again for your valuable feedback. Also, we hope that these responses will help you reassess our work and update your score.

---

### Official Review · Reviewer_MBb9 · 2025-07-02

**Rating:** 4
**Confidence:** 4

**Summary:**

This paper introduces SustainLC, a high-fidelity benchmark environment for testing reinforcement learning methods to control liquid cooling systems in large data centers. Built on a detailed simulation of the Frontier supercomputer’s cooling system, it allows RL agents to optimize energy use while keeping server temperatures safe. The benchmark supports multi-agent setups, centralized and hybrid control, and includes tools for explainable RL using decision trees and large language models. Overall, the work is timely and well-executed.

**Dataset Code Accessibility:**

Yes

**Ethical Considerations:**

No, there are no or only very minor ethics concerns

**Final Justification:**

I appreciate and have read authors' followups and would maintain my current score.

**Limitations Weaknesses:**

- The benchmark is simulation-only using a digital twin; there is no demonstration of real-world deployment or validation against physical systems.

- Only PPO is used throughout the study. Other RL algorithms, such as SAC and TD3, or offline RL approaches, are not evaluated. The authors should show or explain the compatibility among varying RL algorithms.

- The explainability component is a strength, but its evaluation is relatively shallow. There’s little quantification of how interpretability impacts control quality, and decision tree depth (up to 28) may still be hard to audit.

- For a benchmark aimed at explainability, the LLM modules should be benchmarked themselves (e.g., by human evaluation, factual accuracy checks, or alignment with true system dynamics). Without this, the added value of the LLM component is uncertain.

**Strengths Contributions:**

+ Practical: The benchmark tackles a real-world challenge by using RL to manage cooling in data centers like the Frontier supercomputer.

+ Scalability design: It supports large setups with many cooling units by using centralized inference and decomposed observations.

+ Interpretability design: The authors distill trained RL policies into decision trees, allowing users to understand control decisions in a structured way.

+ Explainability: The authors integrate large language models to explain the RL agent's actions in natural language.

---

> ### Author Rebuttal · Authors · 2025-07-30
>
> ### Rebuttal to Reviewer MBb9
>
> We appreciate your thoughtful review and the valuable feedback provided. We acknowledge the limitations you highlighted and would like to address them respectfully while presenting our arguments for consideration.
>
> 1. **Benchmark**.
> The primary goal of this foundational paper is to introduce a sustainability benchmark with an example of a high fidelity design of a real data center at scale. One of the best high fidelity end-to-end liquid cooling model to our knowledge is the Frontier supercomputer digital twin of Oak Ridge National Labs from the ExaDigiT consortium of supercomputer users. This democratizes access to ML researchers and enables a community to contribute to the high-impact problem of sustainability.
>
> 2. **Inclusion of Additional RL Algorithms**:
>     We recognize the importance of evaluating multiple reinforcement learning (RL) algorithms. In response to your feedback, we are actively incorporating results from the Soft Actor-Critic (SAC) algorithm into our study. While we initially focused on Proximal Policy Optimization (PPO) due to its on-policy nature and the simulator's capabilities, we understand that SAC offers advantages in sample efficiency for off-policy learning. Since we had a simulator, we used on-policy PPO for learning as it is robust, widely-used, and serves as a strong and representative baseline for demonstrating SustainLC's capabilities. We aim to present SAC results in our final submission. But here the results are provided with the caveat that they are based on unoptimized versions of the SAC algorithm hyperparameters and architectures due to time constraints. We hope that this addition will enhance the robustness of our findings and provide a more comprehensive evaluation of our benchmark. TD3 is a continuous space RL algorithm and our action space is hybrid, so we couldn’t use it as part of the ablation.
>
>     However, our benchmark is fundamentally algorithm-agnostic. The standard Gymnasium interface ensures out-of-the-box compatibility with the vast majority of modern RL algorithms, inlcuding offline RL approaches. Through the benchmark focus and standard interface, we have enabled the community to perform exactly the kind of comparisons the reviewer suggests. We will clarify this in the final version by explicitly stating the environment's compatibility with other RL frameworks.
>
> ### **FOR SAC**
>
> | Agent Type                | Control Details        | D_blade | ∑P (kW) | ∑Q   | Avg Reward Per Cab | Avg Reward Per CT |
> |--------------------------|------------------------|-------------------|---------|-------|----------------------|---------------------|
> | Baseline                 | ASHRAEG36              | 76.92             | 237.31  | 235.28 | 1697.08              | 360.17              |
> | CTRL+BGBaseline          | OnlyCTRLcontrol        | 44.61             | 208.37  | 197.41 | 1386.94              | 392.81              |
> | CTBaseline+BGRL          | NoValveControl         | 55.05             | 235.16  | 212.98 | 1528.68              | 364.35              |
> | CTBaseline+BGRL          | WithValveControl       | 57.11             | 242.86  | 224.60 | 1531.94              | 358.69              |
> | MultiagentRL             | DecentralizedAction    | 79.31             | 213.84  | 228.35 | 1703.19              | 386.48              |
> | MultiagentRL             | CentralizedAction (CA) | 80.18             | 216.16  | 231.80 | 1704.88              | 381.05              |
> | MultiagentRL             | CA&Multiheadpolicy     | 84.33             | 225.32  | 229.54 | 1709.45              | 375.49              |
>
> 3. **Interpretability of Decision Trees**:
>     We appreciate your concerns regarding the depth of the decision trees used for policy distillation. While a depth of 28 may seem complex, we emphasize that the most critical decisions are derived from the first few splits, which are based on interpretable, physically-grounded variables such as wet-bulb and supply water temperature as shown in Figure 12, which is only up to depth 17. The deeper nodes capture fine-grained optimizations. We have clarified in our text that the primary goal of policy distillation is not merely to replicate the full policy but to uncover high-level control heuristics. This approach allows for future exploration of methods to distill shallower, yet effective, trees, thereby enhancing interpretability without sacrificing performance.
>
> 4. **Evaluation of Explainability**:
>     We acknowledge that the evaluation of the explainability component could be perceived as shallow. To address this, we are  exploring an LLM explainability benchmark where alignment of LLM rationale with system dynamics can be evaluated and ensured. We believe that releasing this benchmark code and the active ExaDigiT community of supercomputer users are also key in coming up with this alignment, as we can scale this effort. We included the LLM component not as a production-ready solution, but as a proof-of-concept to demonstrate a promising future direction. This was one of the purposes this work is a collaboration of HPC orgs at HPE and ORNL. This benchmark has been useful in studying sciencetifically grounded foundation models.
>
> We hope that these clarifications will reinforce the quality of this work and may help reassess the significantly positive impact of this benchmark in democratizing supercomputer optimization for sustainability by the ML community.

---

### Official Review · Reviewer_b2RK · 2025-07-21

**Rating:** 5
**Confidence:** 3

**Summary:**

This paper proposes a benchmark for reinforcement learning-based liquid cooling (LC) control in high performance computing (HPC) systems, called SustainLC. The SustainLC benchmark is built upon digital twins of the cooling system of the Frontier Supercomputer at Oak Ridge National Laboratory and is implemented using a Gymnasium interface. The contribution of the paper is clear, as it addresses the important challenge of optimizing liquid cooling control in HPC systems, which is becoming increasingly relevant as the use of artificial intelligence continues to expand in today’s world. Based on the proposed SustainLC benchmark, the authors demonstrate the advantages of using RL-based control processes over a traditional controller. In addition, the benchmark supports the use of large language models (LLMs) for explaining the RL agent’s control process.

**Dataset Code Accessibility:**

Yes

**Dataset Code Comments:**

The proposed benchmark is available at https://hewlettpackard.github.io/sustain-lc/. It is accessible and seems well-documented.

**Ethical Considerations:**

No, there are no or only very minor ethics concerns

**Final Justification:**

The authors have adequately addressed my concerns regarding the trustworthiness of the LLMs used in their methods, as well as the typographical issues. Therefore, I will maintain my current rating.

**Limitations Weaknesses:**

1. The authors claim that the use of large language models (LLMs) can provide natural language explanations of control actions to foster user trust. However, LLMs can sometimes generate inconsistent explanations. For example, in Section A.4, third example, for the Head 1 output (27.09 °C, 27.87 kg/s), the LLM provides the following explanation: "Coolant flow is modest at 28.11 kg/s, with distribution tilting slightly toward branch 3 (0.35)." The reported coolant flow should be 27.87 kg/s, not 28.11 kg/s. This inconsistency raises concerns about the reliability of LLM-generated explanations. How can we ensure that LLMs do not produce such errors and truly foster user trust? The authors are encouraged to provide a discussion on this issue.
2. There are some typos in the paper. For example, on page 4, "Gymasium" should be corrected to "Gymnasium." On page 9, "Table 6" refers to a figure rather than a table. The caption for Table 5 seems incomplete.

**Strengths Contributions:**

The motivation of this paper is clear and well-justified. It proposes a liquid cooling (LC) control benchmark with the potential to advance reinforcement learning (RL)-based LC control and enhance the overall optimization of the control process. The use of the Gymnasium interface supports both RL-based and traditional control methods. Moreover, the benchmark supports single-agent, multi-agent, homogeneous, and hybrid RL policies within customizable data center setups. Based on the proposed benchmark, the authors show that the multi-agent reinforcement learning (RL) approach with a centralized action (CA) and multi-head policy achieves better performance than both traditional control methods and RL-based control methods without CA and multi-head policy.

---

> ### Author Rebuttal · Authors · 2025-07-30
>
> We would like to express our sincere gratitude to Reviewer b2RK for their thorough review and constructive feedback. We appreciate the time and effort taken to evaluate our submission, and we acknowledge the limitations highlighted in your review.
>
> 1. **Inconsistency**: We are especially grateful for identifying the inconsistency in the coolant flow example (reporting 28.11 kg/s instead of the correct 27.87 kg/s). This error underscores the potential pitfalls of LLMs, which you rightly highlighted. We have highligted this is as a particular example of LLM pitfall.
>
>     To ensure that the LLM-generated explanations accurately reflect the actions predicted by the RL policy, we have employed  two evaluation strategies.
>
>         a. First, we use action reconstruction consistency, where we prompt the LLM with the generated explanation (along with the corresponding observation) and ask it to reconstruct the original action. We then compare the reconstructed action with the RL-predicted action using metrics such as mean absolute error (MAE). A close match between the two indicates that the explanation is faithful and grounded in the decision.
>
>         b. We adopt a human-as-a-judge evaluation approach, where domain-aware annotators assess the explanation based on two criteria:
>             (i) faithfulness — whether the explanation aligns with the given observation and the RL action
>             (ii) completeness — whether the explanation accounts for all relevant dimensions of the action
>             (iii) if the generated explanation is theoretically consistent and factually correct.
>
> Overall, we sincerely appreciate your critical observation regarding the inconsistency in the LLM-generated text explanation in Appendix A.4. To address this important issue completely, we further propose several mitigation strategies that can be explored using our benchmark:
>
> - Post-Hoc Validation: Implementing an automated verification step to parse the LLM's output and cross-reference numerical values against ground-truth data can help flag or correct discrepancies before they are presented to users.
>
> - Retrieval-Augmented Generation (RAG): Grounding the LLM's context with real-time data from the environment can significantly reduce the likelihood of hallucinations.
>
> - Structured Generation : By constraining the LLM to fill predefined templates (e.g., "Coolant flow is set to {flow_rate_value} kg/s."), we can enforce factual consistency for critical parameters.
>
> We included the LLM component not as a production-ready solution but as a proof-of-concept to demonstrate a promising future direction. The error you pointed out emphasizes the necessity of a benchmark like SustainLC, which provides a controlled environment to study and mitigate these failure modes before real-world deployment. This was one of the purposes this work is a collaboration of HPC orgs at HPE and ORNL. This benchmark has been useful in studying sciencetifically grounded foundation models.
>
> 2. **Typographical Errors**: Thank you for bringing the typographical error of "gymnasium" to our attention. We will address these issues in the manuscript to enhance the overall quality of our submission. We will also update the reference from Table 6 to Figure 6. Thank you bringing this to our eyes!
>
> Thank you once again for your valuable feedback.

---

> > ### Comment · Reviewer_b2RK · 2025-08-03
> >
> > I appreciate the authors' response. My concerns have been adequately addressed.

---

### Note · Authors · 2025-08-13

Dear Area Chair,

We thank you for the opportunity to share our closing remarks. "SustainLC" represents a landmark collaboration between a leading supercomputer manufacturer (HPE-Cray) and a premier national laboratory (Oak Ridge). For the first time, a high-fidelity, end-to-end liquid cooling model of a state-of-the-art supercomputer is being released as an open-source benchmark, poised for broad impact through the international ExaDigiT consortium of supercomputing centers and universities from across Europe, the US, and Australia. This is a pivotal contribution that will empower the entire AI community to tackle the critical challenge of sustainable computing through end-to-end liquid cooling optimization in data centers for design and operation refinements.

The paper's contributions, strengthened and clarified through constructive reviewer feedback, mark a significant leap forward:

**Benchmark Modeling:** It delivers a comprehensive, validated model of a large-scale liquid cooling system and outlines a clear, industry-standard sim-to-real pathway, providing the essential, risk-free platform for developing control strategies for next-generation AI infrastructure. This modeling is customizable through the API for broad adoption.

**Advanced Multi-Agent RL Control:** It introduces a sophisticated, multi-objective RL framework critical for managing the immense heat from high-density server cabinets, a vital step for enabling the future of sustainable AI with the increasing power of accelerators.

**Pioneering Agentic AI for Control:** It introduces a paradigm shift by successfully implementing LLM-based reasoning for complex physical control, with additional results showing LLMs can rival RL. It provides the benchmark needed to study the capabilities of LLMs while mitigating their failure modes in control settings, through a reference agentic design. This pioneering work on explainable AI for AI infrastructure optimization has profound implications beyond data centers to other critical sustainability applications.

"SustainLC" is a foundational platform that bridges the gap between industrial-scale systems and cutting-edge AI research. It directly addresses the urgent need for sustainable practices in an era of exponential AI growth. We are confident this paper will inspire a new wave of research and believe its novelty and impact make it an ideal candidate for consideration for oral presentation.

Thank you for your time and consideration.

---

### Decision · Program_Chairs · 2025-09-18

**Decision:**

Accept (poster)

**Comment:**

I think that this is a borderline paper. No reviewers recommended me to reject this paper. At the same time, no reviewers strongly recommended me to accept this paper. Their recommendations are "5. Accept" and "4. Borderline Accept". As a result, the average score is 4.5.

It seems to me that the main strength of this paper is practical usefulness. All reviewers think that this paper has high practical usefulness. This paper proposes a reinforcement learning benchmark platform for optimizing energy efficiency in state-of-the-art liquid-cooled high-performance supercomputers. The model, which is used for testing reinforcement learning methods, is customizable through the API for broad adoption.

Most reviewers pointed out the necessity of evaluating explanations given by LLMs about control actions. In the proposed model, control actions can be explained by LLMs. However, the explanations cannot be evaluated. It is unclear whether the explanations are correct or not. In the authors' rebuttal, this issue is handled by the following comment: "We included the LLM component not as a production-ready solution but as a proof-of-concept to demonstrate a promising future direction."

Most reviewers also pointed out the necessity of validating the proposed digital twin-based model using real-world physical systems. In the authors' rebuttal, the authors explain the high fidelity of the simulation model and the reason for using the digital twin-based model.

Since this is a borderline paper, I think that the final decision will be made by Program Chair. My recommendation is to accept this paper for its practical usefulness and the importance of the application field: Liquid colling control of supercomputers.